# Robust Lipschitz Bandits to Adversarial Corruptions

**Yue Kang**
Department of Statistics
University of California, Davis
Davis, CA 95616
yuekang@ucdavis.edu

**Cho-Jui Hsieh**
Google and Department of Computer Science, UCLA
Los Angeles, CA
chohsieh@cs.ucla.edu

**Thomas C. M. Lee**
Department of Statistics
University of California, Davis
Davis, CA 95616
tcmlee@ucdavis.edu

## Abstract

Lipschitz bandit is a variant of stochastic bandits that deals with a continuous arm set defined on a metric space, where the reward function is subject to a Lipschitz constraint. In this paper, we introduce a new problem of Lipschitz bandits in the presence of adversarial corruptions where an adaptive adversary corrupts the stochastic rewards up to a total budget $C$. The budget is measured by the sum of corruption levels across the time horizon $T$. We consider both weak and strong adversaries, where the weak adversary is unaware of the current action before the attack, while the strong one can observe it. Our work presents the first line of robust Lipschitz bandit algorithms that can achieve sub-linear regret under both types of adversary, even when the total budget of corruption $C$ is unrevealed to the agent. We provide a lower bound under each type of adversary, and show that our algorithm is optimal under the strong case. Finally, we conduct experiments to illustrate the effectiveness of our algorithms against two classic kinds of attacks.

## 1 Introduction

Multi-armed Bandit (MAB) [3] is a fundamental and powerful framework in sequential decision-making problems. Given the potential existence of malicious users in real-world scenarios [10], a recent line of works considers the stochastic bandit problem under adversarial corruptions: an agent adaptively updates its policy to choose an arm from the arm set, and an adversary may contaminate the reward generated from the stochastic bandit before the agent could observe it. To robustify bandit learning algorithms under adversarial corruptions, several algorithms have been developed in the setting of traditional MAB [16, 18, 29] and contextual linear bandits [6, 12, 17, 27, 39]. These works consider either the weak adversary [29], which has access to all past data but not the current action before choosing its attack, or the strong adversary [6], which is also aware of the current action for contamination. Details of these two adversaries will be elaborated in Section 3. In practice, bandits under adversarial corruptions can be used in many real-world problems such as pay-per-click advertising with click fraud and recommendation systems with fake reviews [29], and it has been empirically validated that stochastic MABs are vulnerable to slight corruption [12, 15, 18].

Although there has been extensive research on the adversarial robustness of stochastic bandits, most existing works consider problems with a discrete arm set, such as the traditional MAB and contextual linear bandit. In this paper, we investigate robust bandit algorithms against adversarial corruptions in the Lipschitz bandit setting, where a continuously infinite arm set lie in a known metric space

37th Conference on Neural Information Processing Systems (NeurIPS 2023).

Table 1: Comparisons of regret bounds for our proposed robust Lipschitz bandit algorithms.

| ALGORITHM | REGRET BOUND | FORMAT | $C$ | ADVERSARY |
|---|---|---|---|---|
| Robust Zooming | $\tilde{O}\left(T^{\frac{d_z+1}{d_z+2}} + C^{\frac{1}{d_z+1}}T^{\frac{d_z}{d_z+1}}\right)$ | HIGH. PROB. | KNOWN | STRONG |
| RMEL | $\tilde{O}\left((C^{\frac{1}{d_z+2}}+1)T^{\frac{d_z+1}{d_z+2}}\right)$ | HIGH. PROB. | UNKNOWN | WEAK |
| EXP3.P | $\tilde{O}\left((C^{\frac{1}{d+2}}+1)T^{\frac{d+2}{d+3}}\right)$ | EXPECTED | UNKNOWN | STRONG |
| CORRAL | $\tilde{O}\left((C^{\frac{1}{d+1}}+1)T^{\frac{d+1}{d+2}}\right)$ | EXPECTED | UNKNOWN | STRONG |
| BoB Robust Zooming | $\tilde{O}\left(T^{\frac{d+3}{d+4}} + C^{\frac{1}{d+1}}T^{\frac{d+2}{d+3}}\right)$ | HIGH. PROB. | UNKNOWN | STRONG |

with covering dimension $d$ and the expected reward function is an unknown Lipschitz function. Lipschitz bandit can be used to efficiently model many real-world tasks such as dynamic pricing, auction bidding [35] and hyperparameter tuning [19]. The stochastic Lipschitz bandit has been well understood after a large body of literature [8, 23, 30], and state-of-the-art algorithms could achieve a cumulative regret bound of order $\tilde{O}(T^{\frac{d_z+1}{d_z+2}})$[1] in time $T$. However, to the best of our knowledge, the stochastic Lipschitz bandit problem with adversarial corruptions has never been explored, and we believe it is challenging since most of the existing robust MAB algorithms utilized the idea of elimination, which is much more difficult under a continuously infinite arm pool. Furthermore, the complex structure of different metric spaces also poses challenges for defending against adversarial attacks [36] in theory. Therefore, it remains intriguing to design computationally efficient Lipschitz bandits that are robust to adversarial corruptions under both weak and strong adversaries.

We develop efficient robust algorithms whose regret bounds degrade sub-linearly in terms of the corruption budget $C$. Our contributions can be summarized as follows: (1) Under the weak adversary, we extend the idea in [29] and propose an efficient algorithm named Robust Multi-layer Elimination Lipschitz bandit algorithm (RMEL) that is agnostic to $C$ and attains $\tilde{O}(C^{\frac{1}{d_z+2}}T^{\frac{d_z+1}{d_z+2}})$ regret bound. This bound matches the minimax regret bound of Lipschitz bandits [8, 23] in the absence of corruptions up to logarithmic terms. This algorithm consists of multiple parallel sub-layers with different tolerance against the budget $C$, where each layer adaptively discretizes the action space and eliminates some less promising regions based on its corruption tolerance level in each crafted epoch. Interactions between layers assure the promptness of the elimination process. (2) Under the strong adversary, we first show that when the budget $C$ is given, a simple modification on the classic Zooming algorithm [23] would lead to a robust method, namely, Robust Zooming algorithm, which could obtain a regret bound of order $\tilde{O}(T^{\frac{d_z+1}{d_z+2}} + C^{\frac{1}{d_z+1}}T^{\frac{d_z}{d_z+1}})$. We then provide a lower bound to prove the extra $O(C^{\frac{1}{d_z+1}}T^{\frac{d_z}{d_z+1}})$ regret is unavoidable. Further, inspired by the Bandit-over-Bandit (BoB) model selection idea [11, 13, 31], we design a two-layer framework adapting to the unknown $C$ where a master algorithm in the top layer dynamically tunes the corruption budget for the Robust Zooming algorithm. Three types of master algorithms are discussed and compared in both theory and practice. Table 1 outlines our algorithms as well as their regret bounds under different scenarios.

## 2 Related Work

**Stochastic and Adversarial Bandit** Extensive studies have been conducted on MAB and its variations, including linear bandit [1], matrix bandit [20], etc. The majority of literature can be categorized into two types of models [24]: stochastic bandit, in which rewards for each arm are independently sampled from a fixed distribution, and adversarial bandit, where rewards are maliciously generated at all time. However, adversarial bandit differs from our problem setting in the sense that rewards are arbitrarily chosen without any budget or distribution constraint. Another line of work aims to obtain "the best of both worlds" guarantee simultaneously [7]. However, neither of these models is reliable in practice [9], since the former one is too ideal, while the latter one remains very pessimistic, assuming a fully unconstrained setting. Therefore, it is more natural to consider the scenario that lies "in between" the two extremes: the stochastic bandit under adversarial corruptions.

---

[1]$\tilde{O}$ ignores the polylogarithmic factors. $d_z$ is the zooming dimension defined in Section 3.

**Lipschitz Bandit** Most existing works on the stochastic Lipschitz bandit [2] follow two key ideas. One is to uniformly discretize the action space into a mesh in the initial phase so that any MAB algorithm could be implemented [21, 30]. The other is to adaptively discretize the action space by placing more probes in more promising regions, and then UCB [8, 23, 28], TS [19] or elimination [14] method could be utilized to deal with the exploration-exploitation tradeoff. The adversarial Lipschitz bandit was recently introduced and solved in [32], where the expected reward Lipschitz function is arbitrarily chosen at each round. However, as mentioned in the previous paragraph, this fully adversarial setting is quite different from ours. And their algorithm relies on several unspecified hyperparameters and hence is computationally formidable in practice. In addition, some interesting variations of Lipschitz bandits have also been carefully examined, such as contextual Lipschitz bandits [33], full-feedback Lipschitz bandits [22], and taxonomy bandits [34].

**Robust Bandit to Adversarial Corruptions** Adversarial attacks were studied in the setting of MAB [18] and linear bandits [15]. And we will use two classic attacks for experiments in Section 6. To defend against attacks from weak adversaries, [29] proposed the first MAB algorithm robust to corruptions with a regret $C$ times worse than regret in the stochastic setting. An improved algorithm whose regret only contains an additive term on $C$ was then proposed in [16]. [27] subsequently studied the linear bandits with adversarial corruptions and achieved instance-dependent regret bounds. [25] also studied the corrupted linear bandit problem while assuming the attacks on reward are linear in action. Recently, a robust VOFUL algorithm achieving regret bound only logarithmically dependent on $T$ was proposed in [37]. Another line of work on the robust bandit problem focuses on a more challenging setting with strong adversaries who could observe current actions before attacking rewards. [6] considered the corrupted linear bandit when small random perturbations are applied to context vectors, and [12, 17, 39] extended the OFUL algorithm [1] and achieved improved regret bounds. [38] further considers general non-linear contextual bandits and also MDPs with strong adversarial corruptions. However, the study of Lipschitz bandits under attacks remains an unaddressed open area.

## 3 Preliminaries

We will introduce the setting of Lipschitz bandits with adversarial corruptions in this section. The Lipschitz bandit is defined on a triplet $(\mathcal{X}, D, \mu)$, where $\mathcal{X}$ is the arm set space equipped with some metric $D$, and $\mu : \mathcal{X} \to \mathbb{R}$ is an unknown Lipschitz reward function on the metric space $(\mathcal{X}, D)$ with Lipschitz constant 1. W.l.o.g. we assume $\mathcal{X}$ is compact with its diameter no more than 1. Under the stochastic setting, at each round $t \in [T] := \{1, 2, \ldots, T\}$, stochastic rewards are sampled for each arm $x \in \mathcal{X}$ from some unknown distribution $\mathbb{P}_x$ independently, and then the agent pulls an arm $x_t$ and receives the corresponding stochastic reward $\tilde{y}_t$ such that,

$$\tilde{y}_t = \mu(x_t) + \eta_t, \tag{1}$$

where $\eta_t$ is i.i.d. zero-mean random error with sub-Gaussian parameter $\sigma$ conditional on the filtration $\mathcal{F}_t = \{x_t, x_{t-1}, \eta_{t-1}, \ldots, x_1, \eta_1\}$. W.l.o.g we assume $\sigma = 1$ for simplicity in the rest of our analysis. At each round $t \in [T]$, the weak adversary observes the payoff function $\mu(\cdot)$, the realizations of $\mathbb{P}_x$ for each arm $x \in \mathcal{X}$ and choices of the agent $\{x_i\}_{i=1}^{t-1}$ in previous rounds, and injects an attack $c_t(x_t)$ into the reward before the agent pulls $x_t$. The agent then receives a corrupted reward $y_t = \tilde{y}_t + c_t(x_t)$. The strong adversary would be omniscient and have complete information about the problem $\mathcal{F}_t$. In addition to the knowledge that a weak adversary possesses, it would also be aware of the current action $x_t$ while contaminating the data, and subsequently decide upon the corrupted reward $y_t = \tilde{y}_t + c_t(x_t)$. Some literature in corrupted bandits [12, 15] also consider attacking on the contexts or arms, i.e. the adversary modifies the true arm $x_t$ in a small region, while in our problem setting it is obvious that attacking contexts is only a sub-case of attacking rewards due to the Lipschitzness of $\mu(\cdot)$, and hence studying the adversarial attacks on rewards alone is sufficient under the Lipschitz bandit setting.

The total corruption budget $C$ of the adversary is defined as $C = \sum_{t=1}^{T} \max_{x \in \mathcal{X}} |c_t(x)|$, which is the sum of maximum perturbation from the adversary at each round across the horizon $T$. Note the strong adversary may only corrupt the rewards of pulled arms and hence we could equivalently write $C = \sum_{t=1}^{T} |c_t(x_t)|$ in that case as [6, 17]. Define the optimal arm $x_* = \arg\max_{x \in \mathcal{X}} \mu(x)$ and the loss of arm $x$ as $\Delta(x) = \mu(x_*) - \mu(x), x \in \mathcal{X}$. W.l.o.g. we assume $C \leq T$ and each instance of attack $|c_t(x)| \leq 1, \forall t \in [T], x \in \mathcal{X}$ as in other robust bandit literature [16, 29] since

---

**Algorithm 1** Robust Zooming Algorithm

---

**Input:** Arm metric space $(\mathcal{X}, D)$, time horizon $T$, probability rate $\delta$.
**Initialization:** Active arm set $J = \{\}$, active space $\mathcal{X}_{act} = \mathcal{X}$.
 1: **for** $t = 1$ **to** $T$ **do**
 2:     **if** $f(v) - f(u) \geq r(v) + 2r(u)$ for some pair of active arms $u, v \in J$. **then**
 3:         Set $J = J \setminus \{u\}$ and $\mathcal{X}_{act} = \mathcal{X}_{act} \setminus \mathcal{B}(u, r(u))$.         ▷ Removal
 4:     **end if**
 5:     **if** $\mathcal{X}_{act} \nsubseteq \cup_{v \in J} \mathcal{B}(v, r(v))$ **then**
 6:         Activate and pull some arm $x \notin \cup_{v \in J} \mathcal{B}(v, r(v))$ in $\mathcal{X}_{act}$ such that $x_t = x$, $J = J \cup \{x\}$, and set the components $n(x) = 0$, $f(x) = 0$.     ▷ Activation
 7:     **else**
 8:         Pull $x_t = \arg\max_{v \in J} I(v) = f(v) + 2r(v)$, and break ties arbitrarily.     ▷ Selection
 9:     **end if**
10:     Observe the payoff $y_t$. And update components associated with $x_t$ in the Robust Zooming Algorithm: $n(x_t) = n(x_t) + 1$, $f(x_t) = \left( f(x_t)\left(n(x_t) - 1\right) + y_t \right)/n(x_t)$.
11: **end for**

---

the adversary could already make any arm $x \in \mathcal{X}$ optimal given that $\Delta(x) \leq 1$. (We can assume $|c_t(x)| \leq u, \forall t \in [T], x \in \mathcal{X}$ for any positive constant $u$.) Similar to the stochastic case [21], the goal of the agent is to minimize the cumulative regret defined as:

$$Regret_T = T\mu(x_*) - \sum_{t=1}^{T} \mu(x_t). \tag{2}$$

An important pair of concepts in Lipschitz bandits defined on $(\mathcal{X}, D, \mu)$ are the covering dimension $d$ and the zooming dimension $d_z$. Let $\mathcal{B}(x, r)$ denotes a closed ball centered at $x$ with radius $r$ in $\mathcal{X}$, i.e. $\mathcal{B}(x, r) = \{x' \in \mathcal{X} : D(x, x') \leq r\}$, the $r$-covering number $N_c(r)$ of metric space $(\mathcal{X}, D)$ is defined as the minimal number of balls with radius of no more than $r$ required to cover $\mathcal{X}$. On the contrary, the $r$-zooming number $N_z(r)$ introduced in [23] not only depends on the metric space $(\mathcal{X}, D)$ but also the payoff function $\mu(\cdot)$. It describes the minimal number of balls of radius not more than $r/16$ required to cover the $r$-optimal region defined as $\{x \in \mathcal{X} : \Delta(x) \leq r\}$ [8, 14][2]. Next, we define the covering dimension $d$ (zooming dimension $d_z$) as the smallest $q \geq 0$ such that for every $r \in (0, 1]$ the $r$-covering number $N_c(r)$ ($r$-zooming number $N_z(r)$) can be upper bounded by $\alpha r^{-q}$ for some multiplier $\alpha > 0$ that is free of $r$:

$$d = \min\{q \geq 0 : \exists \alpha > 0, N_c(r) \leq \alpha r^{-q}, \forall r \in (0, 1]\},$$
$$d_z = \min\{q \geq 0 : \exists \alpha > 0, N_z(r) \leq \alpha r^{-q}, \forall r \in (0, 1]\}.$$

It is clear that $0 \leq d_z \leq d$ since the $r$-optimal region is a subset of $\mathcal{X}$. On the other hand, $d_z$ could be much smaller than $d$ in some benign cases. For example, if the payoff function $\mu(\cdot)$ defined on the metric space $(\mathbb{R}^k, \|\cdot\|_2), k \in \mathbb{N}$ is $C^2$-smooth and strongly concave in a neighborhood of the optimal arm $x_*$, then it could be easily verified that $d_z = k/2$ whereas $d = k$. However, $d_z$ is never revealed to the agent as it relies on the underlying function $\mu(\cdot)$, and hence designing an algorithm whose regret bound depends on $d_z$ without knowledge of $d_z$ would be considerably difficult.

## 4 Warm-up: Robust Lipschitz Bandit with Known Budgets

To defend against attacks on Lipschitz bandits, we first consider a simpler case where the agent is aware of the corruption budget $C$. We demonstrate that a slight modification of the classic Zooming algorithm [23] can result in a robust Lipschitz bandit algorithm even under the strong adversary, called the Robust Zooming algorithm, which achieves a regret bound of order $\tilde{O}(T^{\frac{d_z+1}{d_z+2}} + C^{\frac{1}{d_z+1}} T^{\frac{d_z}{d_z+1}})$.

We first introduce some notations of the algorithm: denote $J$ as the active arm set. For each active arm $x \in J$, let $n(x)$ be the number of times arm $x$ has been pulled, $f(x)$ be the corresponding average sample reward, and $r(x)$ be the confidence radius controlling the deviation of the sample average

---

[2]We actually use the near-optimality dimension introduced in [8], where the authors imply the equivalence between this definition and the original zooming dimension proposed in [23].

$f(x)$ from its expectation $\mu(x)$. We also define $\mathcal{B}(x, r(x))$ as the confidence ball of an active arm $x$. In essence, the Zooming algorithm works by focusing on regions that have the potential for higher rewards and allocating fewer probes to less promising regions. The algorithm consists of two phases: in the activation phase, a new arm gets activated if it is not covered by the confidence balls of all active arms. This allows the algorithm to quickly zoom into the regions where arms are frequently pulled due to their encouraging rewards. In the selection phase, the algorithm chooses an arm with the largest value of $f(v) + 2r(v)$ among $J$ based on the UCB methodology.

Our key idea is to enlarge the confidence radius of active arms to account for the known corruption budget $C$. Specifically, we could set the value of $r(x)$ as:

$$r(x) = \sqrt{\frac{4\ln(T) + 2\ln(2/\delta)}{n(x)}} + \frac{C}{n(x)},$$

where the first term accounts for deviation in stochastic rewards and the second term is used to defend the corruptions from the adversary. The robust algorithm is shown in Algorithm 1. In addition to the two phases presented above, our algorithm also conducts a removal procedure at the beginning of each round for better efficiency. This step adaptively removes regions that are likely to yield low rewards with high confidence. Theorem 4.1 provides a regret bound for Algorithm 1.

**Theorem 4.1.** *Given the total corruption budget that is at most $C$, with probability at least $1 - \delta$, the overall regret of Robust Zooming Algorithm (Algorithm 1) can be bounded as:*

$$Regret_T = O\left(\ln(T)^{\frac{1}{d_z+2}} T^{\frac{d_z+1}{d_z+2}} + C^{\frac{1}{d_z+1}} T^{\frac{d_z}{d_z+1}}\right) = \tilde{O}\left(T^{\frac{d_z+1}{d_z+2}} + C^{\frac{1}{d_z+1}} T^{\frac{d_z}{d_z+1}}\right).$$

Furthermore, the following Theorem 4.2 implies that our regret bound attains the lower bound and hence is unimprovable. The detailed proof is given in Appendix A.6.1.

**Theorem 4.2.** *Under the strong adversary with a corruption budget of $C$, for any zooming dimension $d_z \in \mathbb{Z}^+$, there exists an instance for which any algorithm (even one that is aware of $C$) must suffer a regret of order $\Omega(C^{\frac{1}{d_z+1}} T^{\frac{d_z}{d_z+1}})$ with probability at least $0.5$.*

In addition to the lower bound provided in Theorem 4.2, we further propose another lower bound for the strong adversary particularly in the case that $C$ is unknown in the following Theorem 4.3:

**Theorem 4.3.** *For any algorithm, when there is no corruption, we denote $R_T^0$ as the upper bound of cumulative regret in $T$ rounds under our problem setting described in Section 3, i.e. $Regret_T \leq R_T^0$ with high probability, and it holds that $R_T^0 = o(T)$. Then under the strong adversary and unknown attacking budget $C$, there exists a problem instance on which this algorithm will incur linear regret $\Omega(T)$ with probability at least $0.5$, if $C = \Omega(R_T^0/4^{d_z}) = \Omega(R_T^0)$.*

However, there are also some weaknesses to our Algorithm 1. The first weakness is that the algorithm is too conservative and pessimistic in practice since the second term of $r(x)$ would dominate under a large given value of $C$. We could set the second term of $r(x)$ as $\min\{1, C/n(x)\}$ to address this issue and the analysis of Theorem 4.1 will still hold as shown in Appendix A.1 Remark A.5. The second weakness is that it still incurs the same regret bound shown in Theorem 4.1 even if there are actually no corruptions applied. To overcome these problems and to further adapt to the unknown corruption budget $C$, we propose two types of robust algorithms in the following Section 5.

## 5 Robust Lipschitz Bandit with Unknown Budgets

In practice, a decent upper bound of the corruption budget $C$ would never be revealed to the agent, and hence it is important to develop robust Lipschitz bandit algorithms agnostic to the amount of $C$. In this section, we first propose an efficient adaptive-elimination-based approach to deal with the weak adversary, and then we adapt our Algorithm 1 by using several advances of model selection in bandits to defend against the attacks from the strong adversary.

### 5.1 Algorithm for Weak Adversaries

The weak adversary is unaware of the agent's current action before contaminating the stochastic rewards. We introduce an efficient algorithm called Robust Multi-layer Elimination Lipschitz bandit algorithm (RMEL) that is summarized in Algorithm 2. Four core steps are introduced as follows.

---

**Algorithm 2** Robust Multi-layer Elimination Lipschitz Bandit Algorithm (RMEL)

---

**Input:** Arm metric space $(\mathcal{X}, D)$, time horizon $T$, probability rate $\delta$, base parameter $B$.

**Initialization:** Tolerance level $v_l = \ln(4T/\delta)B^{l-1}, m_l = 1, n_l = 0, \mathcal{A}_l = 1/2$-covering of $\mathcal{X}$, $f_{l,A} = n_{l,A} = 0$ for all $A \in \mathcal{A}_l, l \in [l^*]$ where $l^* := \min\{l \in \mathbb{N} : \ln(4T/\delta)B^{l-1} \geq T\}$.

1: **for** $t = 1$ **to** $T$ **do**
2:      Sample layer $l \in [l^*]$ with probability $1/v_l$, with the remaining probability sampling $l = 1$. Find the minimum layer index $l_t \geq l$ such that $\mathcal{A}_{l_t} \neq \emptyset$.        ▷ Layer sampling
3:      Choose $A_t = \arg\min_{A \in \mathcal{A}_{l_t}} n_{l_t, A}$, break ties arbitrary.
4:      Randomly pull an arm $x_t \in A_t$, and observe the payoff $y_t$.
5:      Set $n_{l_t} = n_{l_t} + 1, n_{l_t, A_t} = n_{l_t, A_t} + 1$, and $f_{l_t, A_t} = (f_{l_t, A_t}(n_{l_t, A_t} - 1) + y_t)/n_{l_t, A_t}$.
6:      **if** $n_{l_t} = 6\ln(4T/\delta) \cdot 4^{m_l} \times |\mathcal{A}_{l_t}|$ **then**
7:          Obtain $f_{l_t,*} = \max_{A \in \mathcal{A}_{l_t}} f_{l_t, A}$.
8:          For each $A \in \mathcal{A}_{l_t}$, if $f_{l_t,*} - f_{l_t, A} > 4/2^{m_{l_t}}$, then we eliminate $A$ from $\mathcal{A}_{l_t}$ and all active regions $A'$ from $\mathcal{A}_{l'}$ in the case that $A' \subseteq A, A' \in \mathcal{A}_{l'}, l' < l$.        ▷ Removal
9:          Find $1/2^{m_l+1}$-covering of each remaining $A \in \mathcal{A}_{l_t}$ in the same way as $A$ was partitioned in other layers. Then reload the active region set $\mathcal{A}_{l_t}$ as the collection of these coverings.
10:         Set $n_{l_t} = 0, m_{l_t} = m_{l_t} + 1$. And renew $n_{l_t, A} = f_{l_t, A} = 0, \forall A \in \mathcal{A}_{l_t}$.        ▷ Refresh
11:      **end if**
12: **end for**

---

**Multi-layer Parallel Running:** Inspired by the multi-layer idea used in robust MABs [29], our algorithm consists of multiple sub-layers running in parallel, each with a different tolerance level against corruptions. As shown in Algorithm 2, there are $l^*$ layers and the tolerance level of each layer, denoted as $v_l$, increases geometrically with a ratio of $B$ (a hyperparameter). At each round, a layer $l$ is sampled with probability $1/v_l$, meaning that layers that are more resilient to attacks are less likely to be chosen and thus may make slower progress. This sampling scheme helps mitigate adversarial perturbations across layers by limiting the amount of corruptions distributed to layers whose tolerance levels exceed the unknown budget $C$ to at most $O(\ln(T/\delta))$. For the other low-tolerance layers which may suffer from high volume of attacks, we use the techniques introduced below to rectify them in the guidance of the elimination procedure on robust layers. While we build on the multi-layer idea introduced in [29], our work introduces significant refinements and novelty by extending this approach to continuous and infinitely large arm sets, as demonstrated below.

**Active Region Mechanism:** For each layer $\ell$, our algorithm proceeds in epochs: we initialize the epoch index $m_l = 1$ and construct a $1/2^{m_l}$-covering of $\mathcal{X}$ as the active region set $\mathcal{A}_l$. In addition, we denote $n_l$ as the number of times that layer $l$ has been chosen, and for each active region $A \in \mathcal{A}_l$ we define $n_{l,A}, f_{l,A}$ as the number of times $A$ has been chosen as well as its corresponding average empirical reward respectively. Assume layer $l_t$ is selected at time $t$, then only one active region (denoted as $A_t$) in $\mathcal{A}_{l_t}$ would be played where we arbitrarily pull an arm $x_t \in A_t$ and collect the stochastic payoff $y_t$. For any layer $l$, if each active region in $\mathcal{A}_l$ is played for $6\ln(4T/\delta) \cdot 4^m$ times (i.e. line 6 of Algorithm 2), it will progress to the next epoch after an elimination process that is described below. All components mentioned above that are associated with the layer $l$ will subsequently be refreshed (i.e. line 10 of Algorithm 2).

Although the elimination idea has been used for various bandit algorithms [35], applying it to our problem setting poses significant challenges due to three sources of error: (1) uncertainty in stochastic rewards; (2) unknown corruptions assigned to each layer; (3) approximation bias between the pulled arm and its neighbors in the same active region. We encapsulate our carefully designed elimination procedure that is specified in line 8 of Algorithm 2 from two aspects:

**Within-layer Region Elimination and Discretization:** For any layer $l \in [l^*]$, the within-layer elimination occurs at the end of each epoch as stated above. We obtain the average empirical reward $f_{l,A}$ for all $A \in \mathcal{A}_l$ and then discard regions with unpromising payoffs compared with the optimal one with the maximum estimated reward (i.e. $f_{l,*}$ defined in line 7 of Algorithm 2). We further "zoom in" on the remaining regions of the layer $l$ that yield satisfactory rewards: we divide them into $1/2^{m_l+1}$-covering and then reload $\mathcal{A}_l$ as the collection of these new partitions for the next epoch (line 9 of Algorithm 2) for the layer $l$. In consequence, only regions with nearly optimal rewards would remain and be adaptively discretized in the long run.

**Cross-layer Region Elimination:** While layers are running in parallel, it is essential to facilitate communication among them to prevent less reliable layers from getting trapped in suboptimal regions. In our Algorithm 2, if an active region $A \in \mathcal{A}_l$ is eliminated based on the aforementioned rule, then $A$ will also be discarded in all layers $l' \leq l$. This is because the lower layers are faster whereas more vulnerable and less resilient to malicious attacks, and hence they should learn from the upper trustworthy layers whose tolerance levels surpass $C$ by imitating their elimination decisions.

Note that the elimination step (line 8 of Algorithm 2) could be executed either after each epoch or after each round. The former, which is used in the current Algorithm 2, is computationally simpler as it does not require the entire elimination procedure to be repeated each time. On the other hand, the latter version is more precise as it can identify and eliminate sub-optimal regions more quickly. We defer the pseudocode and more details of the latter version in Appendix A.5 due to the space limit. Another tradeoff lies in the selection of the hyperparameter $B$, which controls the ratio of tolerance levels between adjacent layers. With a larger value of $B$, only fewer layers are required, and hence more samples could be assigned to each layer for better efficiency. But the cumulative regret bound would deteriorate since it's associated with $B$ sub-linearly. The cumulative regret bound is presented in the following Theorem 5.1, with its detailed proof in Appendix A.2.

**Theorem 5.1.** *If the underlying corruption budget is $C$, then with probability at least $1 - \delta$, the overall regret of our RMEL algorithm (Algorithm 2) could be bounded as:*

$$Regret_T = \tilde{O}\left(\left((BC)^{\frac{1}{d_z+2}} + 1\right) T^{\frac{d_z+1}{d_z+2}}\right) = \tilde{O}\left(\left(C^{\frac{1}{d_z+2}} + 1\right) T^{\frac{d_z+1}{d_z+2}}\right).$$

*Proof sketch:* It is highly non-trivial to deduce the regret bound depending on $C$ and $d_z$ in Theorem 5.1 without the knowledge of these two values. We first bound the cumulative regret occurred in the robust layers whose tolerance levels $v_l$ are no less than the unknown $C$ by showing the estimated mean $f_{l,A}$ is close to the underlying ground truth $\mu(x), x \in A$ for all time $t$ and active regions $A$ simultaneously with high probability, i.e. we define the set $\Phi$ as follows and prove $P(\Phi) \geq 1 - 3\delta/4$.

$$\Phi = \left\{ |f_{l,A} - \mu(x)| \leq \frac{1}{2^{m_l}} + \sqrt{\frac{4\ln(T) + 2\ln(4/\delta)}{n_{l,A,t}}} + \frac{\ln(T) + \ln(4/\delta)}{n_{l,A,t}} : \right.$$
$$\left. \forall x \in A, \forall A \in \mathcal{A}_l, \forall l \text{ s.t. } v_l \geq C, \forall t \in [T] \right\}.$$

This concentration result can guarantee that regions with unfavorable rewards will be adaptively eliminated. And we could show that the extra cumulative regret from the other vulnerable layers be controlled by the regret occurred in the robust layers. A detailed proof is presented in Appendix A.2.

Note that if no corruption is actually applied (i.e. $C = 0$), our RMEL algorithm could attain a regret bound of order $\tilde{O}(T^{\frac{d_z+1}{d_z+2}})$ which coincides with the lower bound of stochastic Lipschitz bandits up to logarithmic terms. We further prove a regret lower bound of order $\Omega(C)$ under the weak adversary in Theorem 5.2 with its detailed proof in Appendix A.6.2. Therefore, a compelling open problem is to narrow the regret gap by proposing an algorithm whose regret bound depends on $C$ in another additive term free of $T$ under the weak adversary, like [16] for MABs and [17] for linear bandits.

**Theorem 5.2.** *Under the weak adversary with corruption budget $C$, for any zooming dimension $d_z$, there exists an instance such that any algorithm (even is aware of $C$) must suffer from the regret of order $\Omega(C)$ with probability at least $0.5$.*

## 5.2 Algorithm for Strong Adversaries

In Section 4, we propose the Robust Zooming algorithm to handle the strong adversary given the knowledge of budget $C$ and prove that it achieves the optimal regret bound. However, compared with the known budget $C$ case, defending against strong adversaries naturally becomes more challenging when the agent is unaware of the budget $C$. Motivated by the literature on model selection in bandits, we extend our Robust Zooming algorithm by combining it with different master algorithms to learn and adapt to the unknown $C$ on the fly. We consider two approaches along this line: the first approach uses the master algorithms EXP3.P and CORRAL with the smoothing transformation [31] to deal with unknown $C$, which leads to a promising regret bound but a high computational cost. We then equip Robust Zooming algorithm with the efficient bandit-over-bandit (BoB) idea [11] to adapt to the unknown $C$, leading to a more efficient algorithm with a slightly worse regret bound.

**Model Selection:** When an upper bound on $C$ is known, we propose the Robust Zooming algorithm with regret bound $\tilde{O}(T^{\frac{d_z+1}{d_z+2}} + C^{\frac{1}{d_z+1}}T^{\frac{d_z}{d_z+1}})$ against strong adversaries in Section 4. Therefore, it is natural to consider a decent master algorithm that selects between $\lceil \log_2(T) \rceil$ base algorithms where the $i$-th base algorithm is the Robust Zooming algorithm with corruptions at most $2^i$. As $C \leq T$, there must exist a base algorithm that is at most $2C$-corrupted. Here we choose the stochastic EXP3.P and CORRAL with smooth transformation proposed in [31] as the master algorithm due to the following two reasons with respect to theoretical analysis: (1). our action set $\mathcal{A}$ is fixed and the expected payoff is a function of the chosen arm, which satisfies the restrictive assumptions of this master algorithm (Section 2, [31]); (2). the analysis in [31] still works even the regret bounds of base algorithms contain unknown values, and note the regret bound of our Zooming Robust algorithm depends on the unknown $C$. Based on Theorem 3.2 in [31], the expected cumulative regret of our Robust Zooming algorithm with these two types of master algorithms could be bounded as follows:

**Theorem 5.3.** *When the corruption budget $C$ is unknown, by using our Algorithm 1 with $\{2^i\}_{i=1}^{\lceil \log_2(T) \rceil}$ corruptions as base algorithms and the EXP3.P and CORRAL with smooth transformation [31] as the master algorithm, the expected regret could be upper bounded by*

$$\mathbb{E}(Regret_T) = \begin{cases} \tilde{O}\left((C^{\frac{1}{d+2}}+1)T^{\frac{d+2}{d+3}}\right) & \textit{EXP3.P}, \\ \tilde{O}\left((C^{\frac{1}{d+1}}+1)T^{\frac{d+1}{d+2}}\right) & \textit{CORRAL}. \end{cases}$$

We can observe that the regret bounds given in Theorem 5.3 are consistent with the lower bounds presented in Theorem 4.3. And CORRAL is better under small corruption budgets $C$ (i.e. $C = \tilde{O}(T^{\frac{d+1}{d+3}})$) whereas EXP3.P is superior otherwise. Note that the order of regret relies on $d$ instead of $d_z$ since the unknown $d_z$ couldn't be used as a parameter in practice, and both regret bounds are worse than the lower bound given in Theorem 4.2 for the strong adversary. Another drawback of the above method is that a two-step smoothing procedure is required at each round, which is computationally expensive. Therefore, for better practical efficiency, we propose a simple BoB-based method as follows:

**BoB Robust Zooming:** The BoB idea [11] is a special case of model selection in bandits and aims to adjust some unspecified hyperparameters dynamically in batches. Here we use $\lceil \log_2(T) \rceil$ Robust Zooming algorithms with different corruption levels shown above as base algorithms in the bottom layer and the classic EXP3.P [4] as the top layer. Our method, named BoB Robust Zooming, divides $T$ into $H$ batches of the same length, and in one batch keeps using the same base algorithm that is selected from the top layer at the beginning of this batch. When a batch ends, we refresh the base algorithm and use the normalized accumulated rewards of this batch to update the top layer EXP3.P since the EXP3.P algorithm [4] requires the magnitude of rewards should at most be 1 in default. Specifically, we normalize the cumulative reward at the end of each batch by dividing it with $(2H + \sqrt{2H \log(12T/H\delta)})$ due to the fact that the magnitude of the cumulative reward at each batch would at most be this value with high probability as shown in Lemma A.14 in Appendix A.4. Note that this method is highly efficient since a single update of the EXP3.P algorithm only requires $O(1)$ time complexity, and hence the additional computation from updating EXP3.P is only $O(H)$. Due to space limit, we defer Algorithm 4 to Appendix A.5, and the regret bound is given as follows:

**Theorem 5.4.** *When the corruption budget $C$ is unknown, with probability at least $1 - \delta$, the regret of our BoB Robust Zooming algorithm with $H = T^{(d+2)/(d+4)}$ could be bounded as:*

$$Regret_T = \tilde{O}\left(T^{\frac{d+3}{d+4}} + C^{\frac{1}{d+1}}T^{\frac{d+2}{d+3}}\right).$$

Although we could deduce the more challenging high-probability regret bound for this algorithm, its order is strictly worse than those given in Theorem 5.3. In summary, the BoB Robust Zooming algorithm is more efficient and easier to use in practice, while yielding worse regret bound in theory. However, due to its practical applicability, we will implement this BoB Robust Zooming algorithm in the experiments. It is also noteworthy that we can attain a better regret bound with Algorithm 2 under the weak adversary as shown in Theorem 5.1, which aligns with our expectation since the strong adversary considered here is more malicious and difficult to defend against.

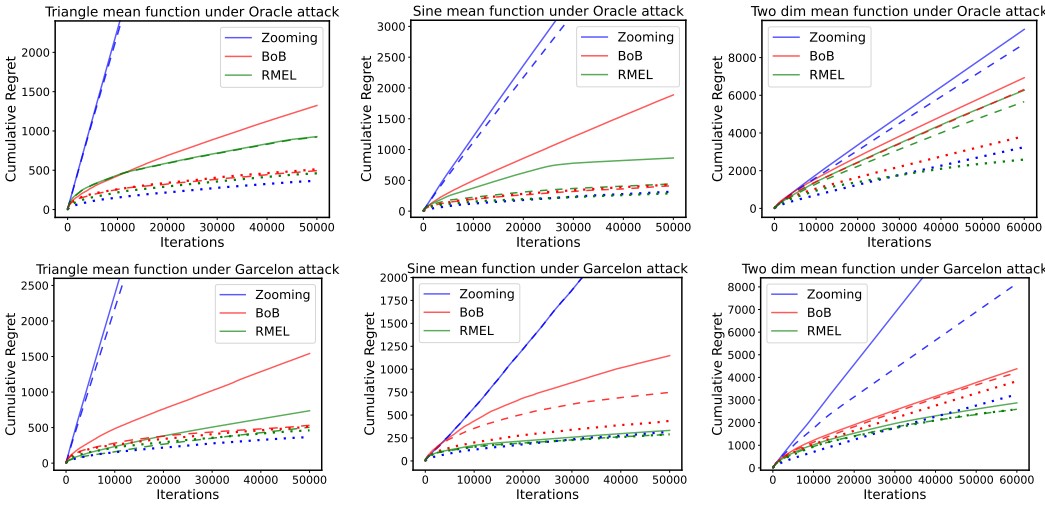

Figure 1: Plots of regrets of Zooming algorithm (blue), RMEL (green) and BoB Robust Zooming algorithm (red) under different settings with three levels of corruptions: (1) dotted line: no corruption; (2) dashed line: moderate corruptions; (3) solid line: strong corruptions. Numerical values of final cumulative regrets in our experiments are also displayed in Table 3 in Appendix A.7.

## 6   Experiments

In this section, we show by simulations that our proposed RMEL and BoB Robust Zooming algorithm outperform the classic Zooming algorithm in the presence of adversarial corruptions. To firmly validate the robustness of our proposed methods, we use three types of models and two sorts of attacks with different corruption levels. We first consider the metric space $([0, 1], |\cdot|)$ with two expected reward functions that behave differently around their maximum: (1). $\mu(x) = 0.9 - 0.95|x - 1/3|$ (triangle) and (2). $\mu(x) = 2/(3\pi) \cdot \sin(3\pi x/2)$ (sine). We then utilize a more complicated metric space $([0, 1]^2, \|\cdot\|_\infty)$ with the expected reward function (3). $\mu(x) = 1 - 0.8\|x - (0.75, 0.75)\|_2 - 0.4\|x - (0, 1)\|_2$ (two dim). We set the time horizon $T = 50,000\,(60,000)$ for the metric space with $d = 1\,(2)$ and the false probability rate $\delta = 0.01$. The random noise at each round is sampled IID from $N(0, 0.01)$. Average cumulative regrets over 20 repetitions are reported in Figure 1.

Since adversarial attacks designed for stochastic Lipschitz bandits have never been studied, we extend two types of classic attacks, named Oracle [18] for the MAB and Garcelon [15] for the linear bandit, to our setting. The details of these two attacks are summarized as follows:

- **Oracle:** This attack [18] was proposed for the traditional MAB, and it pushes the rewards of "good arms" to the very bottom. Specifically, we call an arm is benign if the distance between it and the optimal arm is no larger than 0.2. And we inject this attack by pushing the expected reward of any benign arm below that of the worst arm with an additional margin of 0.1 with probability 0.5.

- **Garcelon:** We modify this type of attack studied in [15] for linear bandit framework, which replaces expected rewards of arms outside some targeted region with IID Gaussian noise. For $d = 1$, since the optimal arm is set to be $1/3$ for both triangle and sine payoff functions, we set the targeted arm interval as $[0.5, 1]$. For $d = 2$, since the optimal arm is close to $(0.75, 0.75)$, we set the targeted region as $[0, 0.5]^2$. Here we contaminate the stochastic reward if the pulled arm is not inside the target region by modifying it into a random Gaussian noise $N(0, 0.01)$ with probability 0.5.

We consider the strong adversary in experiments as both types of attack are injected only if the pulled arms lie in some specific regions. Note although we originally propose RMEL algorithm for the weak adversary in theory, empirically we find it works exceptionally well (Figure 1) across all settings here. We also conduct simulations based on the weak adversary and defer their settings and results to Appendix A.7 due to the limited space. The first Oracle attack is considered to be more malicious in the sense that it specifically focuses on the arms with good rewards, while the second Garcelon attack could corrupt rewards generated from broader regions, which may contain some "bad arms" as well.

Since there is no existing robust Lipschitz bandit algorithm, we use the classic Zooming algorithm [23] as the baseline. As shown in Figure 1, we consider three levels of quantities of corruptions applied on each case to show how attacks progressively disturb different methods. Specifically, we set $C = 0$ for the non-corrupted case, $C = 3,000$ for the moderate-corrupted case and $C = 4,500$ for the strong-corrupted case. Due to space limit, we defer detailed settings of algorithms to Appendix A.7.

From the plots in Figure 1, we observe that our proposed algorithms consistently outperform the Zooming algorithm and achieve sub-linear cumulative regrets under both types of attacks, whereas the Zooming algorithm becomes incompetent and suffers from linear regrets even under a moderate volume of corruption. This fact also implies that the two types of adversarial corruptions used here are severely detrimental to the performance of stochastic Lipschitz bandit algorithms. And it is evident our proposed RMEL yields the most robust results under various scenarios with different volumes of attacks. It is also worth noting that the Zooming algorithm attains promising regrets under a purely stochastic setting, while it experiences a huge increase in regrets after the corruptions emerge. This phenomenon aligns with our expectation and highlights the fact that our proposed algorithms balance the tradeoff between accuracy and robustness in a much smoother fashion.

## 7 Conclusion

In this work we introduce a new problem of Lipschitz bandits in the presence of adversarial corruptions, and we originally provide efficient algorithms against both weak adversaries and strong adversaries when agnostic to the total corruption budget $C$. The robustness and efficiency of our proposed algorithms is then validated under comprehensive experiments.

**Limitations:** For both the weak and strong adversary, our work leaves a regret gap between the lower bound deduced in Theorem 5.2 (Theorem 4.2) and the upper bound in Theorem 5.1 (Theorem 5.3) when agnostic to $C$. Closing or narrowing this regret gap seems highly non-trivial since the regret gap still exists under the simpler MAB setting [16], and we will leave it as a future work.

## Acknowledgments and Disclosure of Funding

We appreciate the insightful comments from the anonymous reviewers and area chair. This work was partially supported by the National Science Foundation under grants CCF-1934568, DMS-1916125, DMS-2113605, DMS-2210388, IIS-2008173 and IIS2048280.

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
