# A    Appendix

## A.1    Analysis of Theorem 4.1

We modify the proof in [23] by dividing the cumulative regret into two parts, where the first part controls the error coming from the stochastic rewards and the second part deals with the extra error from adversarial corruptions in the following Appendix A.1.2. In the beginning we will present some auxiliary lemmas for preparation.

### A.1.1    Useful Lemmas

**Definition A.1.** We call it a clean process for Algorithm 1, if for each time $t \in [T]$ and each active arm $v \in \mathcal{X}$ at any time $t$, we have $|f(v) - \mu(v)| \leq r(v)$.

Here we expand some notations from Algorithm 1: we denote $n_t(v)$ as the number of times the arm $v$ has been pulled until the round $t$, and $f_t(x), r_t(x)$ as the corresponding average stochastic rewards and confidence radius respectively at time $t$ such that,

$$r_t(x) = \sqrt{\frac{4 \ln(T) + 2 \ln(2/\delta)}{n_t(x)}} + \frac{C}{n_t(x)}.$$

Note in our Algorithm 1 we do not write this subscript $t$ for these components since there is no ambiguity in the description. And W.l.o.g we assume the optimal arm $x_* = \arg\max_{x \in \mathcal{X}} \mu(x)$ is unique in $\mathcal{X}$.

**Lemma A.2.** Given the adversarial corruptions are at most $C$, for Algorithm 1, the probability of a clean process is at least $1 - \delta$.

*Proof.* For each time $t \in [T]$, consider an arm $x \in \mathcal{X}$ that is active by the end of time $t$. Recall that when Algorithm 1 pulls the arm $x$, the reward is sampled IID from some unknown distribution $\mathbb{P}_x$ with expectation $\mu(x)$. And in the meanwhile, the stochastic reward may be corrupted by the adversary. Define random variables $U_{x,s}$ and values $C_{x,s}$ for $1 \leq s \leq n_t(x)$ as follows: for $s \leq n_t(x)$, $U_{x,s}$ is the stochastic reward from the $s$-th time arm $x$ is played and $C_{x,s}$ is the corruption injected on $U_{x,s}$ before the agent observing it. By applying Bernstein's Inequality, it naturally holds that

$$P\left(|f_t(x) - \mu(x)| \geq r_t(x)\right) = P\left(|f_t(x) - \mu(x)| \geq \sqrt{\frac{4 \ln T + 2 \ln(2/\delta)}{n_t(x)}} + \frac{C}{n_t(x)}\right)$$

$$= P\left(\left|\sum_{s=1}^{n_t(x)} \frac{U_{x,s}}{n_t(x)} + \sum_{s=1}^{n_t(x)} \frac{C_{x,s}}{n_t(x)} - \mu(x)\right| \geq \sqrt{\frac{4 \ln T + 2 \ln(2/\delta)}{n_t(x)}} + \frac{C}{n_t(x)}\right)$$

$$\leq P\left(\left|\sum_{s=1}^{n_t(x)} \frac{U_{x,s}}{n_t(x)} - \mu(x)\right| + \sum_{s=1}^{n_t(x)} \frac{|C_{x,s}|}{n_t(x)} \geq \sqrt{\frac{4 \ln T + 2 \ln(2/\delta)}{n_t(x)}} + \frac{C}{n_t(x)}\right)$$

$$\overset{(i)}{\leq} P\left(\left|\sum_{s=1}^{n_t(x)} \frac{U_{x,s}}{n_t(x)} - \mu(x)\right| \geq \sqrt{\frac{4 \ln T + 2 \ln(2/\delta)}{n_t(x)}}\right) \leq 2 \cdot \exp\left(-\frac{n_t(x)}{2} \times \frac{4 \ln T + 2 \ln(2/\delta)}{n_t(x)}\right)$$

$$= \delta T^{-2},$$

where the inequality (i) comes from the fact that the total corruption budget is at most $C$. Since there are at most $t$ active arms by time $t$, by taking the union bound over all active arms it holds that,

$$P\left(\forall \text{ active arm } x \text{ at round } t, |f_t(x) - \mu(x)| \leq r_t(x)\right) \geq 1 - \delta T^{-1}, \quad \forall t \in [T].$$

Finally, we take the union bound over all round $t \leq T$, and it holds that,

$$P\left(\forall t \leq T, \forall \text{ active arm } x \text{ at round } t, |f_t(x) - \mu(x)| \leq r_t(x)\right) \geq 1 - \delta T^{-1},$$

which implies that the probability of a clean process is at least $1 - \delta$. $\qquad\square$

**Lemma A.3.** If it is a clean process and the optimal arm $x_* \in B(v, r_t(v))$, then $\mathcal{B}(v, r_t(v))$ could never be eliminated from Algorithm 1 for any $t \in [T]$ and active arm $v$ at round $t$.

*Proof.* Recall that from Algorithm 1, at round $t$ the ball $\mathcal{B}(u, r_t(u))$ would be discarded if we have for some active arm $v$ s.t.

$$f_t(v) - r_t(v) > f_t(u) + 2r_t(u).$$

If $x_* \in \mathcal{B}(u, r_t(u))$, then it holds that

$$f_t(u) + 2r_t(u) \overset{(i)}{\geq} \mu(u) + r_t(u) \geq \mu(u) + D(u, x_*) \overset{(ii)}{\geq} \mu(x_*),$$

where inequality (i) is due to the clean process and inequality (ii) comes from the fact that $\mu(\cdot)$ is a Lipschitz function. On the other hand, we have that for any active arm $v$,

$$\mu(v) \geq f_t(v) - r_t(v), \quad \mu(x_*) \geq \mu(v).$$

Therefore, it naturally holds that

$$f_t(v) - r_t(v) \leq f_t(u) + 2r_t(u).$$

$\square$

**Lemma A.4.** *If it is a clean process, then for any time $t$ and any (previously) active arm $v$ we have $\Delta(v) \leq 3r_t(v)$. Furthermore, we could deduce that $D(u, v) \geq \min\{\Delta(u), \Delta(v)\}/3$ for any pair of (previously) active arms $(u, v)$ by the time horizon $T$.*

*Proof.* Let $S_t$ be the set of all arms that are active or were once active at round $t$. Suppose an arm $x_t$ is played at time $t$. If $x_t$ is just played for one time, i.e. $x_t$ is just activated at time $t$, then we naturally have that,

$$\Delta(x_t) \leq 1 \leq 3r_t(x_t),$$

since the diameter of $\mathcal{X}$ is at most 1. Otherwise, if $x_t$ was played before, i.e. $x_t$ is chosen based on the selection rule instead of the activation rule, we will claim that

$$\mu(x_*) \leq f_t(x_t) + 2r_t(x_t) \leq \mu(x_t) + 3r_t(x_t),$$

under a clean process. First we will show that $f_t(x_t) + 2r_t(x_t) \geq \mu(x_*)$. Recall that the optimal arm $x_*$ is never eliminated according to A.3 under a clean process and hence is covered by some confidence ball, i.e. $x_* \in \mathcal{B}(x', r_t(x')), \exists x' \in S_t$. Then based on the selection rule, it holds that

$$f_t(x_t) + 2r_t(x_t) \geq f_t(x') + 2r_t(x') \geq \mu(x') + r_t(x') \geq \mu(x_*) + r_t(x') - D(x_*, x') \geq \mu(x_*).$$

On the other hand, it holds that,

$$f_t(x_t) + 2r_t(x_t) \leq \mu(x_t) + 3r_t(x_t)$$

since it is a clean process. And these two results directly imply that

$$\mu(x_*) - \mu(x_t) = \Delta(x_t) \leq 3r_t(x_t). \tag{3}$$

For the other active arms $v \in S_t$ that was played before time $t$, let $s < t$ be the last time arm $v$ was played, where we have $f_t(v) = f_s(v)$ and $r_t(v) = r_s(v)$, and then based on Eqn. (3) it holds that $\Delta(v) \leq 3r_s(v) = 3r_t(v)$.

Furthermore, we will show that $D(u, v) \geq \min\{\Delta(u), \Delta(v)\}/3$ for any pair of active arms $(u, v)$ by the time horizon $T$. W.l.o.g we assume that $v$ was activated before $u$, and $u$ was first activated at some time $s'$. Then if $v$ was active at the time $s'$ it naturally holds that $D(u, v) > r_{s'}(v) \geq \Delta(v)/3$ according to the activation rule. If $v$ was removed at the time $s'$ then we also have $D(u, v) > r_{s'}(v)$ since $u$ was not among the discarded region, and hence $D(u, v) \geq \Delta(v)/3$ holds as well. And this concludes our proof. $\square$

### A.1.2 Proof of Theorem 4.1

We modify the original argument for Zooming algorithm [23] to decently resolve the presence of adversarial corruptions. In summary, we could bound the cumulative regret of order $\tilde{O}\left(T^{\frac{d_z+1}{d_z+2}} + C^{\frac{1}{d_z+1}}T^{\frac{d_z}{d_z+1}}\right)$: the first term is the regret caused by the stochastic rewards, which is identical to the regret we have without any corruptions; the second quantity bounds the additional regret caused by the corruptions.

Denote $S_T$ as the active (or previously active) arm set across the time horizon $T$. Then based on Lemma A.4, for any $x \in S_T$ it holds that,

$$\Delta(x) \leq 3r_T(x) = 3\sqrt{\frac{4\ln(T) + 2\ln(2/\delta)}{n_T(x)}} + \frac{3C}{n_T(x)}.$$

And this indicates that

$$\Delta(x)n_T(x) \leq 3\sqrt{\left(4\ln(T) + 2\ln\left(\frac{2}{\delta}\right)\right)n_T(x)} + 3C. \tag{4}$$

Then we denote

$$B_{i,T} = \left\{v \in S_T : 2^i \leq \frac{1}{\Delta(v)} < 2^{i+1}\right\}, \quad \text{where } S_T = \bigcup_{i=0}^{+\infty} B_{i,T},$$

and write $r_i = 2^{-i}$. Then for arbitrary $u, v \in B_{i,T}, i \geq 0$, we have

$$\frac{r_i}{2} < \Delta(u) \leq r_i, \quad \frac{r_i}{2} < \Delta(v) \leq r_i,$$

which implies that $D(x,y) > r_i/6$ under a clean process based on Lemma A.4. Based on the definition of the zooming dimension $d_z$, it follows that $|B_{i,T}| \leq O(r_i^{d_z})$. Subsequently, for any $0 < \rho < 1$ it holds that

$$\sum_{\substack{v \in S_T, \\ \Delta(v) > \rho}} 1 \leq \sum_{i < -\log_2(\rho)} O(r_i^{-d_z}) = O\left(\frac{1}{\rho^{d_z}}\right). \tag{5}$$

Now we define the set $I$ as:

$$I := \left\{v \in S_T : C \leq \sqrt{\left(4\ln(T) + 2\ln\left(\frac{2}{\delta}\right)\right)n_T(v)}\right\}.$$

When an arm $v$ is in the set $I$, the cumulative regret in terms of it would be more related to the stochastic errors other than the adversarial attacks. Subsequently, we could divide the cumulative regret into two quantities:

$$Regret_T = \sum_{v \in S_T} \Delta(v)n_T(v) = \sum_{v \in S_T \cap I} \Delta(v)n_T(v) + \sum_{v \in S_T \cap I^c} \Delta(v)n_T(v)$$

$$= \sum_{\substack{v \in S_T \cap I, \\ \Delta(v) \leq \rho_1}} \Delta(v)n_T(v) + \sum_{\substack{v \in S_T \cap I, \\ \Delta(v) > \rho_1}} \Delta(v)n_T(v) + \sum_{\substack{v \in S_T \cap I^c, \\ \Delta(v) \leq \rho_2}} \Delta(v)n_T(v) + \sum_{\substack{v \in S_T \cap I^c, \\ \Delta(v) > \rho_2}} \Delta(v)n_T(v)$$

$$\overset{(i)}{\leq} \rho_1 T + 2\sum_{\substack{v \in S_T \cap I, \\ \Delta(v) > \rho_1}} 3\sqrt{\left(4\ln(T) + 2\ln\left(\frac{2}{\delta}\right)\right)n_T(v)} + \rho_2 T + 2\sum_{\substack{v \in S_T \cap I^c, \\ \Delta(v) > \rho_2}} 3C$$

$$\overset{(ii)}{\lesssim} \rho_1 T + \sqrt{\ln\left(\frac{T}{\delta}\right)}\sqrt{\left(\sum_{\substack{v \in S_T \cap I, \\ \Delta(v) > \rho_1}} n_T(v)\right)\left(\sum_{\substack{v \in S_T \cap I, \\ \Delta(v) > \rho_1}} 1\right)} + \rho_2 T + C\sum_{\substack{v \in S_T \cap I^c, \\ \Delta(v) > \rho_2}} 1$$

$$\lesssim \rho_1 T + \sqrt{\ln\left(\frac{T}{\delta}\right)}\sqrt{\left(\sum_{\substack{v \in S_T \cap I, \\ \Delta(v) > \rho_1}} n_T(v)\right)\left(\sum_{\substack{v \in S_T \cap I, \\ \Delta(v) > \rho_1}} 1\right)} + \rho_2 T + C\sum_{\substack{v \in S_T \cap I^c, \\ \Delta(v) > \rho_2}} 1$$

$$\overset{(iii)}{\lesssim} \rho_1 T + \sqrt{\ln\left(\frac{T}{\delta}\right)}\sqrt{T}\left(\frac{1}{\rho_1}\right)^{\frac{d_z}{2}} + \rho_2 T + C\left(\frac{1}{\rho_2}\right)^{d_z}. \tag{6}$$

The inequality (i) comes from the definition of set $I$ and Eqn. (4), and inequality (ii) is due to the Cauchy-Schwarz inequality where $\lesssim$ denotes "less in order". Furthermore, we get inequality (iii) based on Eqn. (5). Note Eqn. (6) holds for arbitrary $\rho_1, \rho_2 \in (0, 1)$, and hence by taking

$$\rho_1 = T^{-\frac{1}{d_z+2}} \ln(T)^{\frac{1}{d_z+2}}, \quad \rho_2 = T^{-\frac{1}{d_z+1}} C^{\frac{1}{d_z+1}},$$

we have

$$Regret_T = O\left(\ln(T)^{\frac{1}{d_z+2}} T^{\frac{d_z+1}{d_z+2}} + C^{\frac{1}{d_z+1}} T^{\frac{d_z}{d_z+1}}\right) = \tilde{O}\left(T^{\frac{d_z+1}{d_z+2}} + C^{\frac{1}{d_z+1}} T^{\frac{d_z}{d_z+1}}\right).$$

And this concludes our proof. $\qquad\square$

*Remark* A.5. Note we could replace the second term of $r(x)$ with $\min\{1, C/n(x)\}$, i.e.

$$r_t(x) = \sqrt{\frac{4\ln(T) + 2\ln(2/\delta)}{n_t(x)}} + \min\left\{1, \frac{C}{n_t(x)}\right\},$$

since we know each instance of attack is assumed to be upper bounded by 1. And all our analyses and Lemmas introduced above could be easily verified. Specifically, the core Lemma A.2 still holds as

$$\sum_{s=1}^{n_t(x)} \frac{C_{x,s}}{n_t(x)} \leq \sum_{s=1}^{n_t(x)} \frac{1}{n_t(x)} = 1.$$

## A.2 Analysis of Theorem 5.1

### A.2.1 Useful Lemmas

We first present supportive Lemmas and some of them are adapted from the results in [14, 29].

**Lemma A.6.** *For a sequence of IID Bernoulli trials with a fix success probability $p$, then with probability $1 - \delta$, we could at most observe $[(1 - p)\ln(1/\delta)/p]$ failures until the first success.*

*Proof.* This is based on the property of negative binomial distribution: after we complete the first $N$ trials, the probability of no success is $(1 - p)^N$. To ensure this value is less than $\delta$, we get

$$N = \log_{1-p}(\delta) = \frac{\ln(1/\delta)}{\ln(1/(1-p))} = \frac{\ln(1/\delta)}{\ln(1 + p/(1-p))}.$$

By using the inequality $\ln(x + 1) \leq x, \forall x > -1$, we could take $N = [(1 - p)\ln(1/\delta)/p]$. $\qquad\square$

**Lemma A.7.** (Adapted from Lemma 3.3 [29]) *In Algorithm 2, for any layer whose tolerance level exceeds the unknown $C$, i.e. any layer with index $i \in [l^*]$ s.t. $v_i \geq C$, with probability at least $1 - \delta$, this layer suffers from at most corruptions of amount $(\ln(1/\delta) + 2e - 1)$.*

*Proof.* The proof of this Lemma is an adaptation from the proof of Lemma 3.3 in [29], and we present the detailed proof here for completeness:

In the beginning, we introduce an important result (Lemma 1 in [5]): Let $X_1, \dots, X_T$ be a real-valued martingale difference sequence, i.e. $\forall t \in [T], \mathbb{E}(X_t | X_{t-1}, \dots, X_1) = 0$. And $X_t \leq R$. Denote $V = \sum_{t=1}^T \mathbb{E}(X_t^2 | X_{t-1}, \dots, X_1)$. Then for any $\delta > 0$, it holds that,

$$P\left(\sum_{t=1}^T X_t > R\ln\left(\frac{1}{\delta}\right) + \frac{e - 2}{R} \cdot V\right) \leq \delta.$$

Assume a layer whose tolerance level $\tilde{C}$ is no less than $C$, and hence the probability of pulling this layer would be $1/\tilde{C} \leq 1/C$. For this layer, let $\tilde{C}_x^t$ be the corruption that is observed at round $t$ when arm $x$ is pulled, $x \in X$. Then at any time $t$, if the adversary selects corruption $c_t(a)$ then we know $\tilde{C}_x^t$ is equal to $c_t(a)$ with probability $1/\tilde{C}$ and 0 otherwise. Denote the filtration $\tilde{\mathcal{F}}_t$ containing all the realizations of random variables before time $t$. And hence at time $t$ the adversary could contaminate the stochastic rewards of $\mathcal{X}$ according to $\tilde{\mathcal{F}}_t$. Let $\tilde{a}_t$ be the arm that would be selected if this layer is chosen at the time $t$. Since our Algorithm 2 is deterministic in terms of the active region conditioned

on selecting each layer, and the pulled arm is randomly selected from the active region. Therefore, the selection of $\tilde{a}_t$ is also independent with $\tilde{C}_x^t$ given $\tilde{\mathcal{F}}_t$. We construct the martingale as:

$$X_t = \left|\tilde{C}_x^t\right| - \mathbb{E}\left(\left|\tilde{C}_x^t\right| \mid \tilde{\mathcal{F}}_t\right).$$

Therefore, it holds that

$$\mathbb{E}(X_t^2 | X_{t-1}, \ldots, X_1) = \frac{1}{\tilde{C}}\left(|c_t(a)| - \frac{|c_t(a)|}{\tilde{C}}\right)^2 + \frac{\tilde{C}-1}{\tilde{C}}\left(\frac{|c_t(a)|}{\tilde{C}}\right)^2 \leq 2\frac{|c_t(a)|}{C},$$

since we have that $C \leq \tilde{C}$ and $|c_t(a)| \leq 1$. And conclusively it holds that

$$V = \sum_{t=1}^{T} \mathbb{E}(X_t^2 | X_{t-1}, \ldots, X_1) \leq \sum_{t=1}^{T} 2\frac{|c_t(a)|}{C} \leq 2.$$

Furthermore, it naturally holds that $X_t \leq 1$ due to the fact that $|c_t(a)| \leq 1$. Based on Lemma 1 in [5] we introduced above, with probability at least $1 - \delta$, it holds that

$$\sum_{t=1}^{T} X_t \leq \ln\left(\frac{1}{\delta}\right) + 2(e - 2).$$

On the other hand, we can trivially deduce that the expected corruption injected in this layer is at most 1 since we have total amount of corruptions $C$ and the probability of choosing this layer at each time is fixed as $1/\tilde{C} \leq 1/C$. Conclusively, we have with probability at least $1 - \delta$,

$$\sum_{t=1}^{T} \left|\tilde{C}_x^t\right| = \sum_{t=1}^{T} X_t + \mathbb{E}\left(\sum_{t=1}^{T} \left|\tilde{C}_x^t\right| \mid \tilde{\mathcal{F}}_t\right) \leq \ln\left(\frac{1}{\delta}\right) + 2(e-2) + 1 = \ln\left(\frac{1}{\delta}\right) + 2e - 1.$$

And this completes the proof. $\qquad\qquad\qquad\qquad\qquad\qquad\qquad\qquad\qquad\qquad\qquad\square$

**Definition A.8.** We call it a clean process for Algorithm 2, if for any time $t \in [T]$, any layer $l \in [l^*]$ whose tolerance level $v_l \geq C$, any active region $A \in \mathcal{A}_l$ and any $x \in A$ at time $t$, we have

$$|f_{l,A} - \mu(x)| \leq \frac{1}{2^{m_l}} + \sqrt{\frac{4\ln(T) + 2\ln(4/\delta)}{n_{l,A}}} + \frac{\ln(T) + \ln(4/\delta)}{n_{l,A}}$$

hold for some $0 < \delta < 1$.

To facilitate our analysis in the rest of this section, we expand notations here for Algorithm 2. Similar as in Appendix A.1, we would add the subscript time $t$ to some notations used in Algorithm 2.

- $m_{l,t}$: epoch index of layer $l$ at time $t$;
- $n_{l,t}$: number of selecting the layer $l$ at time $t$ since the last refresh (line 10 of Algorithm 2) on the layer $l$;
- $\mathcal{A}_{l,t}$: active arm set of layer $l$ at time $t$;
- $n_{l,A,t}$: number of selecting the layer $l$ and active region $A \in \mathcal{A}_{l,t}$ by time $t$ since the last refresh on the layer $l$;
- $f_{l,A,t}$: average stochastic rewards of selecting the layer $l$ and active region $A \in \mathcal{A}_{l,t}$ by time $t$ since the last refresh on the layer $l$.

We also denote $l_0$ as the minimum index of layer whose tolerance level just surpasses $C$, i.e. $l_0 = \arg\min\{l \in [l^*] : v_l \geq C\}$. Therefore, we get a clean process defined in Definition A.8 *iff.* the following set $\Phi$ holds:

$$\Phi = \left\{ |f_{l,A,t} - \mu(x)| \leq \frac{1}{2^{m_l}} + \sqrt{\frac{4\ln(T) + 2\ln(4/\delta)}{n_{l,A,t}}} + \frac{\ln(T) + \ln(4/\delta)}{n_{l,A,t}} : \right.$$

$$\left. \forall x \in A, \forall A \in \mathcal{A}_{l,t}, \forall l \in \{l_0, l_0 + 1, \ldots, l^*\}, \forall t \in [T] \right\}. \quad (7)$$

Note the following Lemmas hold for either Algorithm 2 or its variant Algorithm 3 since we define the clean process with respect to each round $t$ instead of each epoch. Note we only need to prove the set $\Phi$ holds at the end of each epoch for the analysis of Algorithm 2. W.l.o.g. we will just prove the regret bound in Theorem 5.1 of Algorithm 2, while it is easy to verify that the same arguments and Theorem 5.1 hold for Algorithm 3 as well.

**Corollary A.9.** *With probability at least $1 - \frac{\delta}{4}$, we select one time of layer $l_0$ at most every $BC \log(4T/\delta)$ times of other layers simultaneously.*

*Proof.* The proof is straight forward based on Lemma A.6. According to the construction of $\{v_l\}_{l=1}^{l^*}$, it holds that $C \leq v_{l_0} < BC$. This implies that the probability of sampling layer $l_0$ at each round is at least $\frac{1}{BC}$. Therefore, after sampling layer $l_0$ in line 2 of Algorithm 2, with probability at least $1 - \frac{\delta}{4T}$, we would sample all the other layers for at most

$$BC \frac{\log(4T/\delta)}{1 - \frac{1}{BC}} \leq BC \log(4T/\delta)$$

times. Since we know the number of time sampling layer $l_0$ is naturally at most $T$, by taking the union bound, we conclude the proof of Corollary A.9. $\square$

**Lemma A.10.** *For algorithm 2, the probability of a clean process is at least $1 - \frac{3}{4}\delta$, i.e. $P(\Phi) \geq 1 - \frac{3}{4}\delta$.*

*Proof.* For each layer $l$ whose tolerance level surpasses $C$, i.e. $l \geq l_0$, we know the probability of sampling this layer in line 2 of Algorithm 2 is at most $1/C$, and this indicates that with probability at least $1 - \delta_1$, this layer suffers from at most $(-\ln(\delta_1) + 2e - 1)$ levels of corruptions based on Lemma A.7. Note the number of layers is less than $\log_B(T)$. This indicates that by taking the union bound on all layers whose tolerance levels surpass $C$, we have with probability at least $1 - \delta_1$, all these layers suffer from at most $\left( \ln\left( \frac{\log_B(T)}{\delta_1} \right) + 2e - 1 \right)$ levels of corruptions across the time horizon $T$. And note

$$\ln\left( \frac{\log_B(T)}{\delta_1} \right) + 2e - 1 \leq \ln\left( \frac{T}{\delta_1} \right)$$

since it is natural to have $T/\log_B(T) \geq e^3$. Then for any time $t$, any layer $l \geq l_0$ and any active region $A \in \mathcal{A}_{l,t}$, define $x_{A,s}$, $C_{A,s}$ and random variables $U_{A,s}$ as the $s$-th time arm pulled, the stochastic reward from pulling $x_{A,s}$ and the corruption injected on $U_{A,s}$ for $1 \leq s \leq n_{l,A,t}$. Also denote

$$r_{l,A,t} = \frac{1}{2^{m_{l,t}}} + \sqrt{\frac{4\ln(T) + 2\ln(4/\delta)}{n_{l,A,t}}} + \frac{\ln(T) + \ln(4/\delta)}{n_{l,A,t}}.$$

With probability at least $1 - \delta/4$, from the above argument, we know that all layers with the index at least $l_0$ suffer from at most $\ln\left( \frac{4T}{\delta} \right)$ levels of corruptions across the time horizon $T$. Denote this event as $\Psi$, i.e. $P(\Psi) \geq 1 - \delta/4$, then under $\Psi$ it holds that

$P\left( |f_{l,A,t} - \mu(x)| \leq r_{l,A,t}, \ \forall x \in A \right)$

$= P\left( \left| \sum_{s=1}^{n_{l,A,t}} \frac{U_{x,s}}{n_{l,A,t}} + \sum_{s=1}^{n_{l,A,t}} \frac{C_{x,s}}{n_{l,A,t}} - \mu(x) \right| \leq r_{l,A,t}, \ \forall x \in A \right)$

$\geq P\left( \left| \sum_{s=1}^{n_{l,A,t}} \frac{U_{x,s}}{n_{l,A,t}} - \sum_{s=1}^{n_{l,A,t}} \frac{\mu(x_{A,s})}{n_{l,A,t}} \right| + \left| \sum_{s=1}^{n_{l,A,t}} \frac{\mu(x_{A,s})}{n_{l,A,t}} - \mu(x) \right| + \left| \sum_{s=1}^{n_{l,A,t}} \frac{C_{x,s}}{n_{l,A,t}} \right| \leq r_{l,A,t}, \forall x \in A \right)$

$\overset{(i)}{\geq} P\left( \left| \sum_{s=1}^{n_{l,A,t}} \frac{U_{x,s}}{n_{l,A,t}} - \sum_{s=1}^{n_{l,A,t}} \frac{\mu(x_{A,s})}{n_{l,A,t}} \right| + \left| \sum_{s=1}^{n_{l,A,t}} \frac{\mu(x_{A,s})}{n_{l,A,t}} - \mu(x) \right| \leq \frac{1}{2^{m_{l,t}}} + \sqrt{\frac{2\ln(4T^2/\delta)}{n_{l,A,t}}}, \forall x \in A \right)$

$\overset{(ii)}{\geq} P\left( \left| \sum_{s=1}^{n_{l,A,t}} \frac{U_{x,s}}{n_{l,A,t}} - \sum_{s=1}^{n_{l,A,t}} \frac{\mu(x_{A,s})}{n_{l,A,t}} \right| \leq \sqrt{\frac{2\ln(4T^2/\delta)}{n_{l,A,t}}} \right)$

$\geq 1 - \frac{\delta}{2} \cdot T^{-2}.$

Inequality (i) is due to the definition of event $\Psi$ and inequality (ii) comes from the fact that the diameter of $A$ is at most $1/2^{m_{l,t}}$ and $\mu(\cdot)$ is a Lipschitz function. We know that at most $T$ active regions would be played across time $T$. By taking the union bound on all rounds $t \in [T]$ and all active regions that have been played, it holds that

$$P\left(|f_{l,A,t} - \mu(x)| \le r_{l,A,t}, \forall x \in A, \forall A \in \mathcal{A}_{l,t}, \forall l \in \{l_0, l_0 + 1, \dots, l^*\}, \forall t \in [T]\right) \ge 1 - \frac{\delta}{2}$$

under the event $\Psi$. Since $P(\Psi) \ge 1 - \delta/4$, overall it holds that

$$P\left(|f_{l,A,t} - \mu(x)| \le r_{l,A,t}, \forall x \in A, \forall A \in \mathcal{A}_{l,t}, \forall l \in \{l_0, l_0 + 1, \dots, l^*\}, \forall t \in [T]\right) \ge 1 - \frac{3\delta}{4},$$

i.e. $P(\Phi) \ge 1 - 3\delta/4$. And this concludes our proof. $\qquad\square$

**Lemma A.11.** *We have $r_{l,A,t} \le 2/2^{m_{l,t}}$ if $n_{l,A,t} = 6\ln(4T/\delta) \cdot 4^{m_{l,t}}$.*

*Proof.* Based on the formulation of $r_{l,A,t}$

$$r_{l,A,t} = \frac{1}{2^{m_{l,t}}} + \sqrt{\frac{4\ln(T) + 2\ln(4/\delta)}{n_{l,A,t}}} + \frac{\ln(T) + \ln(4/\delta)}{n_{l,A,t}}.$$

It suffices to show that

$$\sqrt{\frac{4\ln(T) + 2\ln(4/\delta)}{n_{l,A,t}}} + \frac{\ln(T) + \ln(4/\delta)}{n_{l,A,t}} \le \frac{1}{2^{m_{l,t}}} \tag{8}$$

by taking $n_{l,A,t} = 6\ln(4T/\delta) \cdot 4^{m_{l,t}}$. Firstly, we have that

$$\sqrt{\frac{4\ln(T) + 2\ln(4/\delta)}{n_{l,A,t}}} \le 2\sqrt{\frac{\ln(T) + \ln(4/\delta)}{6\ln(4T/\delta) \cdot 4^{m_{l,t}}}} \le 2\sqrt{\frac{\ln(T) + \ln(4/\delta)}{(3 + 2\sqrt{2})\ln(4T/\delta) \cdot 4^{m_{l,t}}}}$$

$$\le (2\sqrt{2} - 2)\frac{1}{2^{m_{l,t}}}$$

Secondly, it holds that

$$\frac{\ln(T) + \ln(4/\delta)}{n_{l,A,t}} \le \frac{1}{3 + 2\sqrt{2}}\frac{1}{4^{m_{l,t}}} \le (3 - 2\sqrt{2})\frac{1}{2^{m_{l,t}}}.$$

Combining the above two results, we have Eqn. (8) holds, which concludes our proof. $\qquad\square$

**Lemma A.12.** *Under a clean process, for any layer $l$ whose tolerance level $v_l$ is no less than $C$, i.e. $l \ge l_0$, it holds that*

$$\Delta(x) \le 16/2^{m_{l,t}}, \quad \forall x \in A, \forall A \in \mathcal{A}_{l,t}, \forall t \in [T].$$

*Proof.* Here we will focus on Algorithm 2, and the same argument could be used for its variant Algorithm 3. Inspired by the techniques in [14], we will show that under a clean process $\Phi$, the optimal arm $x_*$ would never be eliminated from layers whose tolerance levels are no less than $C$. Obviously, the optimal arm $x_*$ is in the covering when $m_{l,t} = 1$, where the whole arm space $\mathcal{X}$ is covered. Assume the layer $l_t$ reaches the end of epoch $m_{l_t,t}$ at time $t$ (i.e. $m_{l_t,t+1} = m_{l_t,t} + 1$), and the optimal arm $x_*$ is contained in some active region $A_* \in \mathcal{A}_{l_t,t}$. Then under a clean process, for any active region $A_0 \in \mathcal{A}_{l_t,t}$ it holds that,

$$f_{l_t,A_*,t} \ge \mu(x_*) - r_{l_t,A_*,t} \ge \mu(x_*) - 2/2^{m_{l_t,t}} \tag{9}$$
$$f_{l_t,A_0,t} \le \mu(x) + r_{l_t,A_0,t} \le \mu(x) + 2/2^{m_{l_t,t}}, \forall x \in A_0 \tag{10}$$

based on Lemma A.11 since we have $n_{l_t,A,t} = 6\ln(4T/\delta) \cdot 4^{m_{l_t,t}}, \forall A \in \mathcal{A}_{l_t,t}$ in the end of the epoch. And since $\mu(x_*) \ge \mu(x), \forall x \in A_0$, it holds that

$$f_{l_t,A_0,t} - f_{l_t,A_*,t} \le 4/2^{m_{l_t,t}}. \tag{11}$$

This implies that $A_*$ will not be removed. Note the above argument holds for any epoch index and any layer whose corruption level surpasses $C$, and hence the optimal arm $x_*$ would never be eliminated from layers whose tolerance levels are no less than $C$.

To prove Lemma A.12. When $m_{l,t} = 1$, it naturally holds since $\Delta(x) \leq 1 \leq 16/2^1$. Otherwise, let $A_*$ be the covering that contains the optimal arm $x_*$ for layer $l$ in the previous epoch $m_{l,t} - 1$, and according to the above argument it is well defined. And we know $x$ is also alive in the previous epoch, where we denote $A_x$ as the covering that contains $x$ in the previous epoch $m_{l,t} - 1$. Denote $t_0$ as the time the last epoch reaches the end of layer $l$ ($m_{l,t} - 1 = m_{l,t_0}$), and then it holds that

$$\Delta(x) \leq f_{l,A_*,t_0} - f_{l,A_x,t_0} + 2r_{l,A_*,t_0} = f_{l,A_*,t_0} - f_{l,A_x,t_0} + \frac{4}{2^{m_{l,t_0}}} = f_{l,A_*,t_0} - f_{l,A_x,t_0} + \frac{8}{2^{m_{l,t}}}$$

since $r_{l,A_*,t_0} = r_{l,A_x,t_0} = 4/2^{m_{l,t_0}}$ at the end of the epoch $m_{l,t_0}$. On the other hand, since $A_x$ was not eliminated at the end of the epoch $m_{l,t_0}$, based on the same argument used with Eqn. (9), (10), (11), we have that

$$f_{l,A_*,t_0} - f_{l,A_x,t_0} \leq \frac{4}{2^{m_{l,t_0}}} = \frac{8}{2^{m_{l,t}}},$$

and this fact indicates that

$$\Delta(x) \leq \frac{16}{2^{m_{l,t}}}.$$

Note this result holds for any layer whose tolerance level surpasses $C$ and any $t \in [T]$. This implies Lemma A.12 holds conclusively. $\qquad \square$

### A.2.2 Proof of Theorem 5.1

*Proof.* If the corruption budget $C \leq \ln(4T/\delta)$, then all the layers' tolerance levels exceed the unknown $C$, in which case based on Lemma A.10, with probability at least $1 - 3\delta/4$, it holds that $\forall x \in A, \forall A \in \mathcal{A}_{l,t}, \forall l \in [l^*], \forall t \in [T]$

$$|f_{l,A,t} - \mu(x)| \leq \frac{1}{2^{m_l}} + \sqrt{\frac{4\ln(T) + 2\ln(4/\delta)}{n_{l,A,t}}} + \frac{\ln(T) + \ln(4/\delta)}{n_{l,A,t}}.$$

We denote $Regret_T(l)$ as the cumulative regret encountered from the layer $l$ across time $T$, which implies that

$$Regret_T = \sum_{l=1}^{l^*} Regret_T(l).$$

For any fixed layer $l \in [l^*]$, we will then show that $Regret_T(l) = \tilde{O}(T^{\frac{d_z+1}{d_z+2}})$. Based on Lemma A.12, we know that for any layer $l$, any arm played after the epoch $m$ would at most incurs a regret of volume $16/2^m$. Note at epoch $m$, the active arm set consists of $1/2^m$-coverings for some region in $\{x \in \mathcal{X} : \Delta(x) \leq 16/2^m\}$. Therefore, the number of active regions at this epoch $m$ could be upper bounded by $\alpha 2^{d_z m}$ for some constant $\alpha > 0$. And for each active region, we will pull it for exactly $6\ln(4T/\delta) \cdot 4^m$ times in epoch $m$. Therefore, the total regret incurred in the epoch $m$ for any layer would at most be

$$\alpha 2^{d_z m} \times 6\ln(4T/\delta) \cdot 4^m \times 16/2^m = 192\alpha\ln(4T/\delta)2^{(d_z+1)m}.$$

Therefore, the total cumulative regret we experience for any layer $l$ could be upper bounded as:

$$\begin{aligned} Regret_T(l) &\leq \sum_{m=1}^{M} 192\alpha\ln\left(\frac{4T}{\delta}\right)2^{(d_z+1)m} + \frac{8}{2^M}T \\ &\leq 192\alpha\ln\left(\frac{4T}{\delta}\right)\frac{2^{(d_z+1)(M+1)} - 2^{d_z+1}}{2^{d_z+1} - 1} + \frac{8}{2^M}T \\ &\leq 384\alpha\ln\left(\frac{4T}{\delta}\right)2^{(d_z+1)M} + \frac{8}{2^M}T, \end{aligned}$$

where the second term bound the total regret after finishing the epoch $M$. Note we could take $M$ as any integer here, even if the epoch $M$ doesn't exist, our bound still works. By taking $M$ as the closest integer to the value $\left(\ln\left(\frac{T}{48\alpha\ln(4T/\delta)}\right)/[(d_z + 2)\ln(2)]\right)$. It holds that

$$Regret_T(l) \lesssim T^{\frac{d_z+1}{d_z+2}}\ln(T/\delta)^{\frac{1}{d_z+2}}, \qquad \forall l \in [l^*].$$

Therefore, it holds that with probability at least $1 - 3\delta/4 \geq 1 - \delta$,

$$Regret_T = \sum_{l=1}^{l^*} Regret_T(l) \lesssim T^{\frac{d_z+1}{d_z+2}} \ln(T/\delta)^{\frac{1}{d_z+2}} \cdot \log_2(T) = \tilde{O}(T^{\frac{d_z+1}{d_z+2}}).$$

On the other hand, if the corruption budget $C > \ln(4T/\delta)$, then not all the layers could tolerate the unknown total budget level $C$. We denote $l_0$ as the minimum index of the layer that is resilient to $C$ as defined in Eqn. 7. Therefore, we could use the above argument to similarly deduce that:

$$\sum_{l=l_0}^{l^*} Regret_T(l) \lesssim T^{\frac{d_z+1}{d_z+2}} \ln(T/\delta)^{\frac{1}{d_z+2}} \cdot \log_2(T) = \tilde{O}(T^{\frac{d_z+1}{d_z+2}}). \tag{12}$$

For the first $(l_0 - 1)$ layers that are vulnerable to attacks, we could control their regret by using the cross-layer region elimination idea. Specifically, it holds that $v_{l_0} \leq BC$, then based on Corollary A.9, we know that with probability at least $1 - \delta/4$, we select one time of layer $l_0$ at most every $BC \log(4T/\delta)$ times of the first $(l_0 - 1)$ non-robust layers. Since the active regions in a lower-index layer are always a subset of the active regions for the layer with a higher index according to our cross-layer elimination rule in Algorithm 2. We know when the layer $l_0$ stays at the epoch $m$, any arm played in the layer $1, 2, \ldots, l_0$ would at most incurs a regret $16/2^m$. Therefore, when the layer $l_0$ stays in epoch $m$, we have probability at least $1 - 3\delta/4 - \delta/4 = 1 - \delta$, the total regret incurred from the first $l_0$ layers altogether could be bounded as

$$BC \log(4T/\delta) \times \alpha 2^{d_z m} \times 6 \ln(4T/\delta) \cdot 4^m \times 16/2^m = 192 BC\alpha \ln(4T/\delta)^2 2^{(d_z+1)m}.$$

Conclusively, it holds that

$$\begin{aligned}
\sum_{l=1}^{l_0} Regret_T(l) &\leq \sum_{m=1}^{M} 192 BC\alpha \ln\left(\frac{4T}{\delta}\right)^2 2^{(d_z+1)m} + \frac{8}{2^M} T \\
&\leq 192 BC\alpha \ln\left(\frac{4T}{\delta}\right)^2 \frac{2^{(d_z+1)(M+1)} - 2^{d_z+1}}{2^{d_z+1} - 1} + \frac{8}{2^M} T \\
&\leq 384 BC\alpha \ln\left(\frac{4T}{\delta}\right)^2 2^{(d_z+1)M} + \frac{8}{2^M} T,
\end{aligned}$$

for arbitrary $M$. Similarly, then we can simply take $M$ as the closest positive integer to the value

$$\left( \ln\left(\frac{T}{48\alpha BC \ln(4T/\delta)}\right) \right) / [(d_z + 2)\ln(2)],$$

and we have that

$$\sum_{l=1}^{l_0} Regret_T(l) \lesssim T^{\frac{d_z+1}{d_z+2}} \left(BC \ln(T/\delta)^2\right)^{\frac{1}{d_z+2}}. \tag{13}$$

Combine the results from Eqn. 12 and Eqn. 13, with probability at least $1 - \delta$, it holds that

$$Regret_T = \tilde{O}\left(T^{\frac{d_z+1}{d_z+2}} \left(B^{\frac{1}{d_z+2}} C^{\frac{1}{d_z+2}} + 1\right)\right) = \tilde{O}\left(T^{\frac{d_z+1}{d_z+2}} \left(C^{\frac{1}{d_z+2}} + 1\right)\right).$$

And this completes our proof. $\square$

## A.3 Analysis of Theorem 5.3

### A.3.1 Useful Lemmas

**Lemma A.13.** (Part of Theorem 3.2 and 5.3 in [31]) *If the regret of the optimal base algorithm could be bounded by $U_*(T, \delta) = O(c(\delta)T^\alpha)$ for some function $c : \mathbb{R} \to \mathbb{R}$ and constant $\alpha \in [1/2, 1)$, the regret of EXP.P and CORRAL with smoothing transformation as the master algorithms are shown in Table 2:*

The proof of this Lemma involves lots of technical details and is presented in [31] elaborately. And hence we would omit the proof here.

Table 2: Table for Lemma A.13

|  | Known $\alpha$, Unknown $c(\delta)$ |
|---|---|
| EXP3.P | $\tilde{O}\left(T^{\frac{1}{2-\alpha}}c(\delta)\right)$ |
| CORRAL | $\tilde{O}\left(T^{\alpha}c(\delta)^{\frac{1}{\alpha}}\right)$ |

### A.3.2 Proof of Theorem 5.3

*Proof.* The proof of our Theorem 5.3 is based on the above Lemma A.13. According to Theorem 4.1, with probability at least $1/\delta$, the regret bound of our Algorithm 1 could be bounded as

$$Regret_T = \tilde{O}\left(T^{\frac{d_z+1}{d_z+2}} + C^{\frac{1}{d_z+1}}T^{\frac{d_z}{d_z+1}}\right) = \tilde{O}\left(T^{\frac{d_z+1}{d_z+2}} + C^{\frac{1}{d_z+2}}T^{\frac{d_z+1}{d_z+2}}\right).$$

Due to the fact that $d_z$ is upper bounded by $d$ and $C = O(T)$, it further holds that

$$Regret_T = \tilde{O}\left(\left(C^{\frac{1}{d_z+2}}+1\right)T^{\frac{d_z+1}{d_z+2}}\right) = \tilde{O}\left(\left(C^{\frac{1}{d+2}}+1\right)T^{\frac{d+1}{d+2}}\right).$$

Therefore, by taking $\alpha = \frac{d+1}{d+2}$ (known) and $c(\delta) = \left(C^{\frac{1}{d+2}}+1\right)$ (unknown) and plugging them into Lemma A.13, we have that

$$\mathbb{E}(Regret_T) = \begin{cases} \tilde{O}\left((C^{\frac{1}{d+2}}+1)T^{\frac{d+2}{d+3}}\right) & \text{EXP3.P,} \\ \tilde{O}\left((C^{\frac{1}{d+1}}+1)T^{\frac{d+1}{d+2}}\right) & \text{CORRAL.} \end{cases}$$

And this concludes our proof. $\square$

### A.4 Analysis of Theorem 5.4

Under the assumption that the diameter of $\mathcal{X}$ is at most 1, we could also assume that $\mu(x) \in [0,1], \forall x \in X$ due to the Lipschitzness of $\mu(\cdot)$ w.l.o.g. in this section.

#### A.4.1 Useful Lemmas

**Lemma A.14.** *In Algorithm 4, for any batch $i \in \left[\left\lceil\frac{T}{H}\right\rceil\right]$, the sum of stochastic rewards could be bounded as*

$$\left|\sum_{t=(i-1)H+1}^{\min\{iH,T\}} y_t\right| \le 2H + \sqrt{2H\log(\frac{12T}{H\delta})}$$

*simultaneously with probability at least $1 - \delta/3$.*

*Proof.* For arbitrary batch index $i \in \left[\left\lceil\frac{T}{H}\right\rceil\right]$, it holds that

$$P\left(\left|\sum_{t=(i-1)H+1}^{\min\{iH,T\}} y_t\right| \ge 2H + \sqrt{2H\log(\frac{12T}{H\delta})}\right)$$

$$= P\left(\left|\sum_{t=(i-1)H+1}^{\min\{iH,T\}} \mu(x_t) + c_t(x_t) + \eta_t\right| \ge 2H + \sqrt{2H\log(\frac{12T}{H\delta})}\right)$$

$$\le P\left(\sum_{t=(i-1)H+1}^{\min\{iH,T\}} |\mu(x_t)| + \sum_{t=(i-1)H+1}^{\min\{iH,T\}} |c_t(x_t)| + \left|\sum_{t=(i-1)H+1}^{\min\{iH,T\}} \eta_t\right| \ge 2H + \sqrt{2H\log(\frac{12T}{H\delta})}\right)$$

$$\le P\left(\left|\sum_{t=(i-1)H+1}^{\min\{iH,T\}} \eta_t\right| \ge \sqrt{2H\log(\frac{12T}{H\delta})}\right) \overset{(i)}{\le} \frac{H}{6T}\delta.$$

The inequality (i) comes from the fact that $\sum_{t=(i-1)H+1}^{iH} \eta_t$ is sub-Gaussian with parameter $H$. Therefore, by taking a union bound on all $\lceil \frac{T}{H} \rceil$ batches, it holds that

$$P\left(\forall i \in \left\lceil \frac{T}{H} \right\rceil : \left| \sum_{t=(i-1)H+1}^{\min\{iH,T\}} y_t \right| \leq 2H + \sqrt{2H \log(\frac{12T}{H\delta})} \right) \geq 1 - \frac{\delta}{3}.$$

And this concludes the proof of Lemma A.14. $\qquad\square$

### A.4.2 Proof of Theorem 5.4

*Proof.* We are ready to prove Theorem 5.4 now. Since we have $\lceil \log_2(T) \rceil$ base algorithms where the $i$-th base algorithm is our Robust Zooming Algorithm (Algorithm 1) with tolerance level $2^i$, we can denote the base algorithm set as $W = \{2^i\}_{i=1}^{\lceil \log_2(T) \rceil}$ in terms of their tolerance levels. For any round $t \in [T]$, let $w_t$ denote the base algorithm chosen from $W$. And denote $x_t(w)$, $w \in W$ as the arm pulled if the base algorithm $w$ is chosen in the beginning of its batch. In other words, we have $x_t = x_t(w_t)$. Denote $C_i$ as the total budget of corruptions in the $i$-th batch and hence $C = \sum_{i=1}^{\lceil T/H \rceil} C_i$, where recall that $C$ is the unknown total budget of corruptions. And we also write $C_* = \max_i C_i$ as the maximum budget in a single batch. Let $w_*$ be the element in $W$ such that $C_* \leq w_* < 2C_*$. Therefore, we could decompose the cumulative regret into the following two quantities:

$$Regret_T = \underbrace{\sum_{t=1}^{T} (\mu(x_*) - \mu(x_t(w_*)))}_{\text{Quantity (I)}} + \underbrace{\sum_{t=1}^{T} (\mu(x_t(w_*)) - \mu(x_t(w_t)))}_{\text{Quantity (II)}} \qquad (14)$$

And it suffices to bound these two quantities respectively. We know the Quantity (I) could be further represented as

$$\sum_{t=1}^{T} (\mu(x_*) - \mu(x_t(w_*))) = \sum_{i=1}^{\lceil \frac{T}{H} \rceil} \sum_{t=(i-1)H+1}^{\min\{iH,T\}} (\mu(x_*) - \mu(x_t(w_*))).$$

Here we will use the results from Theorem 4.1. Note by setting the probability rate as $\delta/3$ in Algorithm 1, we can prove that we have a clean process with probability at least $1 - \delta/3$ (line 5 in Algorithm 4). Although we run the Algorithm 1 here in a batch fashion and the total rounds is $T$, we can still easily show that with probability at least $1 - \delta/3$ we have a clean process for all batches. This is because the proof of Lemma A.2 only relies on taking a union bound over all rounds $T$ where whether a restart is proceeded doesn't matter at all. According to Theorem 4.1 and the choice of $w_*$, the cumulative regret of each batch could be upper bounded by the order of

$$\tilde{O}\left(H^{\frac{d_z+1}{d_z+2}} + C_*^{\frac{1}{d_z+1}} H^{\frac{d_z}{d_z+1}}\right) = \tilde{O}\left(H^{\frac{d_z+1}{d_z+2}} + C_*^{\frac{1}{d+1}} H^{\frac{d}{d+1}}\right),$$

since $C_* \leq H$ naturally holds by definition. Therefore, it holds that

$$\text{Quantity (I)} = \tilde{O}\left(\left\lceil \frac{T}{H} \right\rceil \left(H^{\frac{d_z+1}{d_z+2}} + C_*^{\frac{1}{d+1}} H^{\frac{d}{d+1}}\right)\right) = \tilde{O}\left(TH^{\frac{-1}{d_z+2}} + TC_*^{\frac{1}{d+1}} H^{\frac{-1}{d+1}}\right), \quad (15)$$

with probability at least $1 - \delta/3$. For Quantity (II), according to Lemma A.14, for any batch $i \in \lceil \lceil \frac{T}{H} \rceil \rceil$ the sum of stochastic rewards could be bounded by

$$\left| \sum_{t=(i-1)H+1}^{\min\{iH,T\}} y_t \right| \leq 2H + \sqrt{2H \log\left(\frac{12T}{H\delta}\right)}$$

simultaneously with probability at least $1 - \delta/3$. We denote the event $\Omega$ as

$$\Omega = \left\{ \forall i \in \left\lceil \frac{T}{H} \right\rceil : \left| \sum_{t=(i-1)H+1}^{\min\{iH,T\}} y_t \right| \leq 2H + \sqrt{2H \log\left(\frac{12T}{H\delta}\right)} \right\},$$

---

**Algorithm 3** Alternative Robust Multi-layer Elimination Lipschitz Bandit Algorithm (RMEL)

---

**Input:** Arm metric space $(\mathcal{X}, D)$, time horizon $T$, probability rate $\delta$, base parameter $B$.

**Initialization:** Tolerance level $v_l = \ln(4T/\delta)B^{l-1}, m_l = 1, n_l = 0, \mathcal{A}_l = 1/2$-covering of $\mathcal{X}$, $f_{l,A} = n_{l,A} = 0$ for all $A \in \mathcal{A}_l, l \in [l^*]$ where $l^* := \min\{l \in \mathbb{N} : \ln(4T/\delta)B^{l-1} \geq T\}$.

1: **for** $t = 1$ **to** $T$ **do**
2:     Sample layer $l \in [l^*]$ with probability $1/v_l$, with the remaining probability sampling $l = 1$. Find the minimum layer index $l_t \geq l$ such that $\mathcal{A}_{l_t} \neq \emptyset$.       ▷ Layer sampling
3:     Choose $A_t = \arg\min_{A \in \mathcal{A}_{l_t}} n_{l_t, A}$, break ties arbitrary.
4:     Randomly pull an arm $x_t \in A_t$, and observe the payoff $y_t$.
5:     Set $n_{l_t} = n_{l_t} + 1$, $n_{l_t, A_t} = n_{l_t, A_t} + 1$, and $f_{l_t, A_t} = (f_{l_t, A_t}(n_{l_t, A_t} - 1) + y_t)/n_{l_t, A_t}$.
6:     Obtain $f_{l_t, *} = \max_{A \in \mathcal{A}_{l_t}} f_{l_t, A}$, $n_{l_t, *} = \min_{A \in \mathcal{A}_{l_t}} n_{l_t, A}$.
7:     For each $A \in \mathcal{A}_{l_t}$, if $f_{l_t, *} - f_{l_t, A} > 2/2^{m_{l_t}} + \sqrt{8 \ln(4T^2/\delta)/n_{l_t, *}} + 2\ln(4T/\delta)/n_{l_t, *}$, then we eliminate $A$ from $\mathcal{A}_{l_t}$ and all active regions $A'$ from $\mathcal{A}_{l'}$ in the case that $A' \subseteq A, A' \in \mathcal{A}_{l'}, l' < l$.       ▷ Removal
8:     **if** $n_{l_t} = 6\ln(4T/\delta) \cdot 4^{m_l} \times |\mathcal{A}_{l_t}|$ **then**
9:         Find $1/2^{m_l+1}$-covering of each $A \in \mathcal{A}_{l_t}$ in the same way as $A$ was partitioned in other layers. Then reload the active region set $\mathcal{A}_{l_t}$ as the collection of these coverings.
10:         Set $n_{l_t} = 0$, $m_{l_t} = m_{l_t} + 1$. And renew $n_{l_t, A} = f_{l_t, A} = 0, \forall A \in \mathcal{A}_{l_t}$.       ▷ Refresh
11:     **end if**
12: **end for**

---

and it holds that $P(\Omega) \geq 1 - \delta/3$. And under the event $\Omega$, from Theorem 6.3 in [4], we know with probability at least $1 - \delta/3$, it holds that

$$\text{Quantity (II)} = \tilde{O}\left(\sqrt{H^2 \frac{T}{H}}\right) = \tilde{O}\left(\sqrt{TH}\right). \tag{16}$$

Specifically, in the statement of Theorem 6.3 [4], we have $K = \lceil \log_2(T) \rceil, \delta = \delta/3, T = \lceil \frac{T}{H} \rceil$ here. And we multiply the regret bound in Theorem 6.3 [4] by $\left(2H + \sqrt{2H\log(\frac{12T}{H\delta})}\right)$ as well since the original EXP3.P algorithm requires the magnitude of rewards not exceeding 1. Conclusively, by combining the results from Eqn. 15 and Eqn. 16 and taking a union bound on the probability rates, with probability at least $1 - \delta/3 - \delta/3 - \delta/3 = 1 - \delta$, we have that

$$Regret_T = \tilde{O}\left(TH^{\frac{-1}{d_z+2}} + TC_*^{\frac{1}{d+1}} H^{\frac{-1}{d+1}} + \sqrt{TH}\right).$$

By taking $H = T^{\frac{d+2}{d+4}}$ and using the fact that $d_z \leq d$, it holds that

$$Regret_T = \tilde{O}\left(T^{\frac{d+3}{d+4}} + C_*^{\frac{1}{d+1}} T^{\frac{d^2+4d+2}{(d+1)(d+4)}}\right) = \tilde{O}\left(T^{\frac{d+3}{d+4}} + C_*^{\frac{1}{d+1}} T^{\frac{d+2}{d+3}}\right)$$

$$= \tilde{O}\left(T^{\frac{d+3}{d+4}} + C^{\frac{1}{d+1}} T^{\frac{d+2}{d+3}}\right),$$

with probability at least $1 - \delta$. $\qquad\qquad\qquad\qquad\qquad\qquad\qquad\qquad\qquad\qquad\qquad\square$

## A.5 Additional Algorithms

### A.5.1 Alternative Algorithm for RMEL

Here we present another version of the RMEL algorithm in 3. Instead of executing the elimination process after each epoch of any layer as in 2, here we conduct the elimination at each round. This modification will make our algorithm more accurate and discard less promising regions in a timely manner but will lead to higher computational complexity as well. Note the regret bound of Theorem 5.1 naturally holds since we could use the identical proof to reach the same regret bound here. And we also add an explanation before Corollary A.9 in Appendix A.2.

---

**Algorithm 4** BoB Robust Zooming Algorithm

---

**Input:** Arm metric space $(\mathcal{X}, D)$, time horizon $T$, probability rate $\delta$, batch size $H$.

**Initialization:** Budget set for base algorithms $I = \{2^i\}_{i=1}^N$, $N = \lceil \log_2(T) \rceil$, $\alpha = 2\sqrt{\ln\left(\frac{3NT}{\delta}\right)}$, $\gamma = \min\left\{ \frac{3}{5}, 2\sqrt{\frac{3N\ln(N)}{5T}} \right\}$, weight $w_i = 1, i \in [N]$, cumulative sum $s = 0$.

1: **for** $t = 1$ **to** $T$ **do**
2:     **if** $t \in \{kH + 1 : k \in \mathbb{N}\}$ **then**
3:         For $i = 1, \ldots, N$ set

$$p_i = (1 - \gamma)\frac{w_i}{\sum_{j=1}^N w_j} + \frac{\gamma}{N}.$$

4:         Choose the base algorithm index $i'$ randomly with probability $\{p_i\}_{i=1}^N$.
5:         Refresh the chosen Robust Zooming algorithm (Algorithm 1 with $C = 2^{i'}$) with active arm set $J = \{\}$, active space $\mathcal{X}_{act} = \mathcal{X}$ and probability rate $\delta/3$.
6:     **end if**
7:     Run the chosen Robust Zooming algorithm and receive the reward $y_t$.
8:     Update the chosen Robust Zooming algorithm according to Algorithm 1 and set $s = s + y_t$.
9:     **if** $t \in \{kH : k \in \mathbb{N}^+\}$ **then**
10:        Let $s = s/\left[ p_{i'}\left(2H + \sqrt{2H\log(\frac{12T}{H\delta})}\right)\right]$.
11:        Update EXP3.P component for index $i'$: $w_{i'} = w_{i'} \exp\left(\frac{\gamma}{3N}\left(s + \frac{\alpha}{p_{i'}\sqrt{NT}}\right)\right)$.
12:     **end if**
13: **end for**

---

### A.5.2 BoB Robust Zooming Algorithm

Due to the space limit, we defer the pseudocode of BoB Robust Zooming algorithm here in Algorithm 4. We can observe that the top layer is an EXP3.P algorithm, which chooses the corruption level used for Robust Zooming algorithm in each batch adaptively. For each batch, we run our Robust Zooming algorithm with the chosen corruption level from the top layer, and use the accumulative rewards collected in each batch to update the components of EXP3.P (i.e. line 10 of Algorithm 4). Note we normalize the cumulative reward by dividing it with $(2H + \sqrt{2H\log(\frac{12T}{H\delta})})$, and this is because that we could prove that the magnitude of the cumulative reward at each batch would be at most $(2H + \sqrt{2H\log(\frac{12T}{H\delta})})$ with high probability as shown in Lemma A.14. And the EXP3.P algorithm [4] requires the magnitude of reward should at most be 1 with our chosen values of $\alpha$ and $\gamma$. The regret bound of Algorithm 4 is given in Theorem 5.4 of our main paper.

### A.6 Discussion on Lower Bounds

We now propose Theorem 4.2 and Theorem 5.2 with their detailed proof in Section A.6.1 and Section A.6.2 respectively, where we provide a pair of lower bounds for the strong adversary and the weak adversary.

### A.6.1 Lower Bound for Strong Adversaries

We repeat our Theorem 4.2 for reference here and then provide a detailed proof as follows:

**Theorem 4.2** *Under the strong adversary with corruption budget $C$, for any zooming dimension $d_z \in \mathbb{Z}^+$, there exists an instance such that any algorithm (even is aware of $C$) must suffer from the regret of order $\Omega\left(C^{\frac{1}{d_z+1}} T^{\frac{d_z}{d_z+1}}\right)$ with probability at least $0.5$.*

*Proof.* Here we consider the metric space $([0, 1)^d, l_\infty)$. For arbitrary $\epsilon \in (0, \frac{1}{2})$, we can equally divide the space $[0, 1]^d$ into $1/\epsilon^d$ small $l_\infty$ balls whose diameters are equal to $\epsilon$ by discretizing each axis. (W.l.o.g we assume 1 is divisible by $\epsilon$ for simplicity since otherwise we could take $\lfloor 1/\epsilon^d \rfloor$

instead.) For example, if $d = 2$ and $\epsilon = \frac{1}{2}$, then we can divide the space into $2^2 = 4$ $l_\infty$ balls: $[0, 0.5)^2, [0, 0.5) \times [0.5, 1), [0.5, 1) \times [0, 0.5), [0.5, 1)^2$. We denote these balls as $\{A_i\}_{i=1}^{1/\epsilon^d}, [0, 1)^d = \cup_{i=1}^{1/\epsilon^d} A_i$ and their centers as $\{c_i\}_{i=1}^{1/\epsilon^d}$. (e.g. the center of $[0, 0.5)^2$ is $(0.25, 0.25)$.) Subsequently, we could define a set of functions $\{f_i(\cdot)\}_{i=1}^{1/\epsilon^d}$ as

$$f_i(x) = \begin{cases} \frac{\epsilon}{2} - \|x - c_i\|_\infty, & x \in A_i; \\ 0, & x \notin A_i. \end{cases}$$

We can easily verify that $f_i(\cdot)$ is a 1-Lipschitz function. For the zooming dimension, if $\epsilon$ is of constant scale, then the zooming dimension will become 0. However, in our analysis here, we would let $\epsilon$ rely on $T$ and be sufficiently small so that the zooming dimension is $d$. If the underlying expected reward function is $f_k(\cdot)$ and there is no random noise, consider the strong adversary that shifts the reward of the arm down to whenever the pulled arm is in $A_k$ and doesn't attack the reward otherwise. This attack could be done for roughly $\lfloor C/\epsilon \rfloor$ times. Intuitively, the learner can do no better than pull each arm in $[0, 1]^d$ uniformly. This implies that roughly the learner should do $\lfloor C/\epsilon \rfloor \lfloor 1/\epsilon^d \rfloor$ rounds of uniform exploration before the attack budget $C$ is used up, where the learner pulls arms outside $A_k$ for approximately $\lfloor C/\epsilon \rfloor \cdot \lfloor (1 - \epsilon^d)/\epsilon^d \rfloor$ times. Take $\epsilon = \left(\frac{C}{T}\right)^{\frac{1}{d+1}}$, we know that roughly the learner should do $\lfloor C/\epsilon \rfloor \lfloor 1/\epsilon^d \rfloor = T$ rounds of uniform exploration, and the cumulative regret is at least

$$\left\lfloor \frac{C}{\epsilon} \right\rfloor \cdot \left\lfloor \frac{(1 - \epsilon^d)}{\epsilon^d} \right\rfloor \cdot \epsilon = \Theta\left(C^{\frac{1}{d+1}} T^{\frac{d}{d+1}}\right) = \Theta\left(C^{\frac{1}{d_z+1}} T^{\frac{d_z}{d_z+1}}\right).$$

For a more rigorous argument, note that for the $k$-th instance $f_k(\cdot)$, the adversary could maliciously replace the reward with 0 until the arm in $A_k$ is pulled at least $\lfloor C/\epsilon \rfloor$ times. After $\lfloor C/\epsilon \rfloor \lfloor 1/2\epsilon^d \rfloor$ rounds, for any algorithm even with the information of value $C$, there must be at least $\lfloor 1/(2\epsilon^d) \rfloor$ balls among $\{A_i\}_{i=1}^{1/\epsilon^d}$ that have been pulled for at most $\lfloor C/\epsilon \rfloor$ times. As a consequence, when we choose the problem instance $k$ among these $\lfloor 1/(2\epsilon^d) \rfloor$ balls and set $\epsilon = \left(\frac{C}{T}\right)^{\frac{1}{d+1}}$, then we know that the regret of order

$$\epsilon \cdot \left\lfloor \frac{C}{\epsilon} \right\rfloor \cdot \left(\left\lfloor \frac{1}{2\epsilon^d} \right\rfloor - 1\right) = \Theta\left(C^{\frac{1}{d_z+1}} T^{\frac{d_z}{d_z+1}}\right)$$

is unavoidable. This implies that the regret could be no worse than $\Omega(C^{\frac{1}{d_z+1}} T^{\frac{d_z}{d_z+1}})$ under the strong adversary with probability 0.5. $\square$

For the stochastic Lipschitz bandit problem, based on [33] we know for any algorithm there exists one problem instance such that the expected regret is at least

$$\inf_{r_0 \in (0, 1)} \left(r_0 T + C \log(T) \sum_{r=2^{-i}:i \in \mathbb{N}, r \geq r_0} \frac{N_z(r)}{r}\right),$$

where $N_z(r)$ is the zooming number. And hence the corruption-free lower bound $O\left(\ln(T)^{\frac{1}{d_z+2}} T^{\frac{d_z+1}{d_z+2}}\right)$ is optimal in terms of the zooming dimension $d_z$. Combining this result with our Theorem 4.2, we can conclude that for any algorithm, there exists a corrupted bandit instance where the algorithm must incur $\Omega\left(\max\left\{\ln(T)^{\frac{1}{d_z+2}} T^{\frac{d_z+1}{d_z+2}}, C^{\frac{1}{d_z+1}} T^{\frac{d_z}{d_z+1}}\right\}\right)$ cumulative regret, which coincides with the order of regret for our Robust Zooming algorithm. Conclusively, our algorithm obtains the optimal order of regret under the strong adversary.

We then restate our Theorem 4.3 for reference and then provide a detailed proof:

**Theorem 4.3** *For any algorithm, when there is no corruption, we denote $R_T^0$ as the upper bound of cumulative regret in $T$ rounds under our problem setting described in Section 3, i.e. $Regret_T \leq R_T^0$ with high probability, and it holds that $R_T^0 = o(T)$. Then under the strong adversary and unknown attacking budget $C$, there exists a problem instance on which this algorithm will incur linear regret $\Omega(T)$ with probability at least 0.5, if $C = \Omega(R_T^0/4^{d_z}) = \Omega(R_T^0)$.*

*Proof.* For the case that $d_z = 0$, we consider the metric space $([0,1], l_2)$ and define the Lipschitz function $f_1(\cdot)$ as                                                                    □

$$f_1(x) = \begin{cases} 0.25 - |x - 0.25|, & x \in [0, 0.5]; \\ 0, & x \in (0.5, 1]. \end{cases},$$

and we assume there is no random noise and no adversarial corruption. (We call this instance $I_0$.) For any algorithm with $\mathbb{E}(Regret_T) \leq R_T^0$ when there is no adversarial corruption, we know that

$$\mathbb{E}(\# \text{ iterations playing arms in } (0.5, 1]) \times 0.25 \leq \mathbb{E}(Regret_T) \leq R_T^0,$$

and hence $\mathbb{E}(\# \text{ iterations playing arms in } (0.5, 1]) \leq 4R_T^0$. By Markov inequality, with probability at least 0.5, the number of iterations that play arms in $(0.5, 1]$ is no more than $8R_T^0$.

Next, we define a new problem setting in the same metric space as:

$$f_2(x) = \begin{cases} 0.25 - |x - 0.25|, & x \in [0, 0.5]; \\ x - 0.5, & x \in (0.5, 1]. \end{cases}.$$

And under the setting of $f_2(\cdot)$ there is a malicious strong adversary with budget $C = 4R_T^0$ to attack using the following strategy: whenever the arm in $(0.5, 1]$ is selected and the corruption budget has not been used up, the adversary moves the reward to 0. We call this instance $I_1$. Therefore, before the budget is used up, each selection of arm in $(0.5, 1]$ returns a reward 0, and hence the agent can never tell the difference between $I_0$ and $I_1$ and would follow the same strategy under $I_0$ until the total corruption level reaches $C = 4R_T^0$ and then the adversary stops to contaminate the rewards. And this requires at least $C/0.5 = 2C = 8R_T^0$ rounds in which the agent chooses arms in $(0.5, 1]$. Therefore, with probability of at least 0.5, the regret in $T$ rounds is at least $(T - 8R_T^0)/4 = \Omega(T)$.

For $d_z > 0$, we use the metric space $([0,1]^d, \|\cdot\|_\infty)$ with $d = \lceil 2d_z \rceil$. We first partition the $d$-dimensional cube $[0,1]^d$ into $2^d$ sub-cubes with side length 0.5, i.e. equally divide the cube $[0,1]^d$ into 0.5-radius $l_\infty$ balls whose diameters are equal to 0.5 by discretizing each axis. We denote these balls as $A_i{}_{i=1}^{2^d}$ and the center of these balls as $c_i{}_{i=1}^{2^d}$, e.g. $c_1 = [0.25]^d$. And we denote the vertex of each ball that matches the vertexes of $[0,1]^d$ as $v_i{}_{i=1}^{2^d}$, e.g. $v_1 = [0]^d$. Subsequently, we could define the function $f_1(\cdot)$ as

$$f_1(x) = \begin{cases} 4^{\frac{-d}{d-d_z}} - \|x - c_1\|_\infty^{\frac{d}{d-d_z}}, & x \in A_1; \\ 0, & x \notin A_1. \end{cases}$$

and we assume there is no random noise and no adversarial corruption. (We call this instance $I_0$.) Since the regret of the algorithm under no corruption satisfies that $\mathbb{E}(Regret_T) \leq R_T^0$, and we know that pulling any arm outside $A_1$ will incur a single regret of $4^{\frac{-d}{d-d_z}}$, and hence we have that

$$\mathbb{E}(\# \text{ iterations playing arms not in } A_1) \leq R_T^0 \cdot 4^{\frac{-d}{d-d_z}}.$$

Then by the pigeonhole principle, there exists a sub-ball $2 \leq i \leq 2^d$ such that the expected number of iterations to pull arms in $A_i$ is no more than $R_T^0 \cdot 4^{\frac{-d}{d-d_z}}/(2^d - 1)$. Without loss of generality, we assume $i = 2$, where $c_2 = [0.75, 0.25, \ldots, 0.25]$ and $v_2 = [1, 0, \ldots, 0]$. Similarly by using Markov Inequality, with probability at least 0.5, the number of iterations that play arms in $A_2$ is no more than $2R_T^0 \cdot 4^{\frac{-d}{d-d_z}}/(2^d - 1)$.

Next, we define a new problem setting in the same metric space as:

$$f_2(x) = \begin{cases} 4^{\frac{-d}{d-d_z}} - \|x - c_1\|_\infty^{\frac{d}{d-d_z}}, & x \in A_1; \\ 2^{\frac{-d}{d-d_z}} - \|x - v_2\|_\infty^{\frac{d}{d-d_z}}, & x \in A_2; \\ 0, & x \notin A_1 \cup A_2. \end{cases}.$$

And under the setting of $f_2(\cdot)$ there is a malicious strong adversary with budget $C = 2R_T^0 \cdot 2^{\frac{-d}{d-d_z}}/(2^d - 1) = \Theta(R_T^0/2^d)$ to attack the rewards. (Note $1 \leq d/(d - d_z) \leq 2$). Specifically, the adversary uses the following strategy: whenever the arm in $A_2$ is selected and the corruption budget has not been used up, the adversary moves the reward to 0. We call this instance $I_1$. Therefore, before the budget is used up, each selection of arm in $A_2$ returns a reward 0, and hence the agent can never

tell the difference between $I_0$ and $I_1$ and would follow the same strategy under $I_0$ until the total corruption level reaches $C = 2R_T^0 \cdot 2^{\frac{-d}{d-d_z}}/(2^d - 1)$, and then the adversary stops to contaminate the rewards. And this requires at least $C/2^{\frac{-d}{d-d_z}} = 2R_T^0 \cdot 4^{\frac{-d}{d-d_z}}/(2^d - 1)$ rounds in which the agent chooses arms in $A_2$. Therefore, with probability of at least $0.5$, the regret in $T$ rounds is at least

$$\left(T - \frac{24^{\frac{-d}{d-d_z}} R_T^0}{2^d - 1}\right) \times \left(2^{\frac{-d}{d-d_z}} - 4^{\frac{-d}{d-d_z}}\right) \geq \frac{3}{16}\left(T - \frac{32R_T^0}{2^d - 1}\right) = \Omega(T).$$

$\square$

### A.6.2 Lower Bound for Weak Adversaries

Recall Theorem 5.2 in our main paper:

**Theorem 5.2** *Under the weak adversary with corruption budget $C$, for any zooming dimension $d_z$, there exists an instance such that any algorithm (even is aware of $C$) must suffer from the regret of order $\Omega(C)$ with probability at least $0.5$.*

*Proof.* We can modify the argument of the previous subsection A.6.1 to validate Theorem 5.2. If $d_z = 0$, we could simply use the metric space $([0,1), l_2)$ and the reward function

$$\mu_1(x) = \begin{cases} \frac{1}{2} - |x - \frac{1}{4}|, & x \in [0, 0.5); \\ 0, & x \in [0.5, 1). \end{cases} \qquad \mu_2(x) = \begin{cases} 0, & x \in [0, 0.5); \\ \frac{1}{2} - |x - \frac{3}{4}|, & x \in [0.5, 1). \end{cases}$$

We can easily verify that the zooming dimension $d_z = 0$ holds. Assume there is no random noise, and at each iteration the weak adversary pushes the reward everywhere in $[0, 1)$ to $0$, which would use a $0.5$ budget. Therefore, this attack could last for the first $\lfloor 2C \rfloor$ rounds, when the agent would just receive a $0$ reward regardless of the pulled arm. For any algorithm, it would at least spend for $\lfloor C \rfloor$ rounds on either $[0, 0.5)$ or $[0.5, 1)$ with probability at least $0.5$. By considering the above two reward functions, we know that it would incur $\Omega(C)$ regret with probability at least $0.5$.

For $d_z > 0$, we set $d = \lceil 2d_z \rceil$ and consider the metric space $([0,1]^d, l_\infty)$. Similarly, we can equally divide the space $[0,1]^d$ into $1/2$ small $l_\infty$ balls whose diameters are equal to $1/2$ by discretizing each axis. We denote these balls as $\{A_i\}_{i=1}^{2^d}$, $[0,1)^d = \cup_{i=1}^{2^d} A_i$ and their centers as $\{c_i\}_{i=1}^{2^d}$. (e.g. the center of $[0, 0.5)^2$ is $(0.25, 0.25)$.) Subsequently, we could define a set of functions $\{f_i(\cdot)\}_{i=1}^{1/2^d}$ as

$$\mu_i(x) = \begin{cases} 4^{\frac{-d}{d-d_z}} - \|x - c_i\|_\infty^{\frac{d_z}{d-d_z}}, & x \in A_i; \\ 0, & x \notin A_i. \end{cases}$$

We can easily verify that the zooming dimension of any instance is $d_z$. Assume there is no random noise, and at each iteration, the weak adversary pushes the reward everywhere in $[0, 1)$ to $0$, which would use a $4^{\frac{-d}{d-d_z}}$ budget. Therefore, this attack could last for the first $\lfloor 4^{\frac{d}{d-d_z}} C \rfloor$ rounds, when the agent would just receive a $0$ reward regardless of the pulled arm. After $\lfloor 4^{\frac{d_z}{d-d_z}} C \rfloor$ rounds, for any algorithm even with the information of value $C$, there must be at least $\lfloor 2^{d-1} \rfloor$ balls among $\{A_i\}_{i=1}^{2^d}$ that have been pulled for at most $\lfloor 2^{(\frac{2d}{d-d_z} - d)} C \rfloor = \Theta(C)$ times. As a consequence, as for the problem instance $k$ among these $\lfloor 2^{d-1} \rfloor$ balls, the regret incurred $\Omega(C)$. Similarly, this means that any algorithm must incur $\Omega(C)$ regret with probability $0.5$. $\square$

### A.7 Additional Experimental Details

Note in our main paper we assume that $\sigma = 1$, and our pseudocodes of Algorithms are based on this assumption. When we know a better upper bound for $\sigma$, we could easily modify the components in each algorithm based on $\sigma$. For example, we could modify the confidence radius of any active arm $x$ in Algorithm 1 as

$$r(x) = \sigma\sqrt{\frac{4\ln(T) + 2\ln(2/\delta)}{n(x)}} + \frac{C}{n(x)}.$$

Next, we exhibit the setup of algorithms involved in our experiments as follows:

- **Zooming algorithm [23]:** We use the same setting for stochastic Lipschitz bandit as in [23], and set the radius for each arm as:

$$r(x) = \sigma \sqrt{\frac{4 \ln (T) + 2 \ln (2/\delta)}{n(x)}}.$$

And its implementation is available with the library [26].

- **RMEL (ours):** We use the same parameter setting for RMEL as shown in our Algorithm 2. And based on the experimental results in Figure 1, this method apparently works best under different kinds of attacks and reward functions.

- **BoB Robust Zooming algorithm (ours):** We use the same parameter setting with $\sigma$ for BoB Robust Zooming algorithm as shown in Algorithm 4 without restarting the algorithm after each batch since we found that restarting will sometimes abandon useful information empirically. This BoB-based approach also works well according to Figure 1.

The numerical results of final cumulative regrets in our simulations in Section 6 (Figure 1) are displayed in Table 3.

Note our RMEL (Algorithm 2) is designed to defend against the weak adversary in the theoretical analysis, and hence to be consistent, we also consider the weak adversary for both types of attacks under the same experimental setting and three levels of corrupted budgets. Recall that in the previous experiments in Section 6, the adversary will contaminate the stochastic rewards only if the pulled action is in the specific region (Oracle: benign arm, Garcelon: targeted arm region), and otherwise the adversary will not spend its budget. And hence it is a strong adversary whose action relies on the current arm. To adapt these two attacks into a weak-adversary version, we could simply inject both sorts of attacks at each round based on their principles at each round: the Oracle will uniformly push the expected rewards of all "good arms" below the expected reward of the worst arm with an additional margin of $0.1$ with probability $0.5$ at the very beginning of each round. And the Garcelon will modify the expected rewards of all arms outside the targeted region into a random Gaussian noise $N(0, 0.01)$ with probability $0.5$ ahead of the agent's action. Consequently, adversaries may consume the corruption budget at each round regardless of the pulled arm, and we expect that they will run out of their total budget in fewer iterations than the strong adversary does. We use the same experimental settings as in Section 6, and the results are exhibited in Table 4.

From Table 4, we can see the experimental results under the weak adversary are consistent with those under the strong adversary. The state-of-the-art Zooming algorithm is evidently vulnerable to the corruptions, while our proposed algorithms, especially RMEL, could yield robust performance across multiple settings consistently. We can also observe that compared with the strong adversary, the weak adversary is less malicious than expected.

Another remark is that the adversarial settings used in our experiments may not be consistent with the assumption that $|c_t(x)| \leq 1$, while we find that (1). by modifying the original attacks and restricting the attack volume to be at most one with truncation, we can get a very similar result as shown in Table 3 and Table 4. (2). actually we can change the assumption to $|c_t(x)| \leq u$ where $u$ is an arbitrary positive constant for the theoretical analysis.

| | Algorithm | Budget ($C$) | Oracle | Garcelon |
|---|---|---|---|---|
| | | 0 | 366.58 | 366.58 |
| | Zooming | 3000 | 10883.51 | 10660.17 |
| | | 4500 | 11153.78 | 11487.59 |
| Triangle reward function | | 0 | 512.46 | 512.46 |
| | RMEL | 3000 | 921.95 | 504.78 |
| | | 4500 | 928.27 | 1542.17 |
| | | 0 | 461.16 | 461.16 |
| | BoB Robust Zooming | 3000 | 495.06 | 531.37 |
| | | 4500 | 1323.97 | 736.85 |
| | Algorithm | Budget ($C$) | Oracle | Garcelon |
| | | 0 | 315.94 | 315.94 |
| | Zooming | 3000 | 5289.65 | 3174.26 |
| | | 4500 | 5720.30 | 3174.29 |
| Sine reward function | | 0 | 289.86 | 289.86 |
| | RMEL | 3000 | 442.66 | 289.29 |
| | | 4500 | 862.90 | 332.71 |
| | | 0 | 435.44 | 435.44 |
| | BoB Robust Zooming | 3000 | 414.54 | 746.96 |
| | | 4500 | 1887.35 | 1148.09 |
| | Algorithm | Budget ($C$) | Oracle | Garcelon |
| | | 0 | 3248.54 | 3248.54 |
| | Zooming | 3000 | 8730.73 | 8149.79 |
| | | 4500 | 9496.83 | 13672.00 |
| Two dim reward function | | 0 | 2589.32 | 2589.32 |
| | RMEL | 3000 | 5660.10 | 2590.77 |
| | | 4500 | 6265.09 | 2872.64 |
| | | 0 | 3831.94 | 3831.94 |
| | BoB Robust Zooming | 3000 | 6310.29 | 4217.74 |
| | | 4500 | 6932.09 | 4380.19 |

Table 3: Numerical values of final cumulative regrets of different algorithms under the experimental settings used in Figure 1 in Section 6 (strong adversaries).

| | Algorithm | Budget ($C$) | Oracle | Garcelon |
|---|---|---|---|---|
| Triangle reward function | Zooming | 0 | 366.58 | 366.58 |
| | | 3000 | 10861.72 | 10660.18 |
| | | 4500 | 10862.75 | 10661.99 |
| | RMEL | 0 | 512.46 | 512.46 |
| | | 3000 | 624.29 | 620.96 |
| | | 4500 | 623.50 | 634.59 |
| | BoB Robust Zooming | 0 | 461.16 | 461.16 |
| | | 3000 | 545.27 | 561.77 |
| | | 4500 | 552.66 | 569.51 |
| | Algorithm | Budget ($C$) | Oracle | Garcelon |
| Sine reward function | Zooming | 0 | 315.94 | 315.94 |
| | | 3000 | 5178.81 | 2636.73 |
| | | 4500 | 5186.28 | 2799.22 |
| | RMEL | 0 | 289.86 | 289.86 |
| | | 3000 | 280.62 | 277.22 |
| | | 4500 | 284.94 | 288.06 |
| | BoB Robust Zooming | 0 | 435.44 | 435.44 |
| | | 3000 | 450.21 | 439.08 |
| | | 4500 | 461.13 | 456.36 |
| | Algorithm | Budget ($C$) | Oracle | Garcelon |
| Two dim reward function | Zooming | 0 | 3248.54 | 3248.54 |
| | | 3000 | 6380.37 | 6517.29 |
| | | 4500 | 6991.41 | 6854.05 |
| | RMEL | 0 | 2589.32 | 2589.32 |
| | | 3000 | 3198.06 | 2940.93 |
| | | 4500 | 4231.88 | 4067.16 |
| | BoB Robust Zooming | 0 | 3831.94 | 3831.94 |
| | | 3000 | 4019.08 | 3335.67 |
| | | 4500 | 4901.20 | 4054.05 |

Table 4: Numerical values of final cumulative regrets of different algorithms under the experimental settings introduced in Appendix A.7 (weak adversaries).