# OpenReview forum: "Robust Lipschitz Bandits to Adversarial Corruptions"
_NeurIPS.cc/2023/Conference — NeurIPS 2023 poster_

### Official Review · Reviewer_4cuU · 2023-07-03

**Soundness:** 3 good
**Presentation:** 2 fair
**Contribution:** 2 fair
**Rating:** 5
**Confidence:** 3

**Summary:**

This work studies Lipschitz bandits robust to adversarial corruption, where both strong and weak adversaries are considered. Robust Zooming along with its cumulative regret upper bound is proposed for strong adversary with known corruption budget. For unknown corruption budget, this work proposes RMEL for weak adversary and a two-layer framework BoB for strong adversary. The lower bounds for both adversaries are also proposed, showing Robust Zooming is theoretically optimal for strong adversary. Experimental results are provided to show the practical performance of RMEL and BoB.

**Strengths:**

- This is the first work that studies robust Lipschitz bandit.
- This work considers multiple settings in terms of adversary type and whether the corruption budget is known.
- Both algorithmic upper bounds and lower bounds are provided.

**Weaknesses:**

- For the case of unknown budget, there is a gap between the upper and lower bounds for both weak and strong adversaries.
- In the experiment section, although RMEL and BoB outperform the baseline, their cumulative regrets do not converge (have large slope) in several settings.


**Questions:**

- This work assumed Lipschitz constant one and $\Delta(x)\le 1$. I wonder how the cumulative regret would scale when we have a higher  Lipschitz constant and a higher function magnitude.
- In figure 1, even the 'no corruption' trials have positive simple regret near T. Does it imply that all the algorithms fail to find the optimal arm?
- I'm more interested in the practical performance of Robust Zooming, as its cumulative regret upper bound is already theoretically optimal.

**Limitations:**

The limitations have been discussed.

---

> ### Author Rebuttal · Authors · 2023-08-08
>
> Thank you for your insightful comments on our work. We are happy to know you think our contributions are solid in theory. Please see our response to your concerns as follows:
>
> **Weakness 1: There is a gap between upper and lower bounds for both weak and strong adversaries:**
>
> We also listed this point as a limitation of our work in the Conclusion section. As we mentioned, even for the stochastic multi-armed problem under the weak adversary and unknown corruption budget, a regret gap still exists in terms of $C$ [16]. (The lower bound is $C$ while BARBAR could only attain regret of order $kC$ where $k$ is the number of arms.) Since the Lipschitz bandit is a much more complicated setting with an infinite number of arms $k = +\infty$, it is very challenging to fully close the gap for the more difficult Lipschitz bandit problem.
>
> **Weakness 2: Do not converge in several settings:**
>
> Thank you very much for your careful review of our results. We cut off the iterations early in Figure 1 since the gaps between different algorithms are often clear. For the triangle mean and sine mean, we can clearly see the fast regret explosion of the Zooming algorithm at the early stage. For the two dim mean, we can already see the better performance of our proposed algorithms at iteration 60,000, and hence we just cut off the time.
>
> We re-ran three algorithms with the most challenging two dim mean reward functions until 140,000 rounds, and listed the cumulative regret of algorithms at iterations 80,000,100,000,120,000 and 140,000. Table 7-8 in the PDF in the above Global Rebuttal display the long-time results of used algorithms under each attack. We can observe that convergence of regrets of our algorithms in the long run, and the Zooming algorithm suffers from linear cumulative regret in the end.
>
> **Question 1: I wonder how the cumulative regret would scale when we have a higher Lipschitz constant and a higher function magnitude:**
>
> In our work, we assume the Lipschitz constant is $1$ since it is a common and default assumption under the Lipschitz bandit literature [23]. Given the Lipscthiz constant $U$, we could simply divide the observed rewards by $U$ at each round before using the Lipschitz bandit algorithm, and then the final regret bound of algorithms would be multiplied by the constant U, which implies that the order of regret bounds doesn’t change.
>
> The magnitude of reward functions doesn’t affect the cumulative regret and the implementation of our algorithms. Note the cumulative regret could be represented as $Regret_T = T \Delta(x_t)$. And $\Delta(x), x \in X$ is independent with the magnitude of the Lipschitz reward function $\mu(\cdot)$ since the function values will be uniformly large under a higher function magnitude. For example, $\Delta(x), x \in [-1,1]$ will be the same for $\mu_1(x) = 1 - |x|$ and $\mu_2(x) = 1,000-|x|$.
>
> **Question 2: even the 'no corruption' trials have positive simple regret near T. Does it imply that all the algorithms fail to find the optimal arm?**
>
> 1. Our problem setting is under the Lipscthiz bandit with infinitely many arms, and hence it is impossible to detect the single best arm. We can just guarantee that all the pulled arms in the end are very close to the optimal one.
> 2. All the Lipschitz bandit algorithms choose arms with some randomness, e.g. Zooming algorithm randomly activates new arms to pull, and our RMEL randomly selects one layer at each round to use. This randomness is used to control the exploration-exploitation tradeoff and can guarantee the efficiency of algorithms even under the worst problem instance. But they will naturally lead to minor regret even in the end.
> 3. Under our problem setting, the stochastic rewards are mixed with random noise and malicious attacks, and hence it is inevitable to expect some little perturbation in the end.
>
> **Question 3: I'm more interested in the practical performance of Robust Zooming**
>
> Although the Robust Zooming algorithm obtains the minimax optimal regret bound, we didn’t include it in experiments since it requires the value of $C$ but in reality this value would never be revealed to the agent. During the rebuttal period, we implemented the Robust Zooming algorithm under the identical six settings of Figure 1. And for the second term of $r(x)$ in this algorithm, we used $\min${$1, C/n(x)$} as suggested in line 166 for better practical efficiency. The Table 9-11 in the PDF in the above Global Rebuttal display the performance of the Robust Zooming algorithm against the original Zooming algorithm under the same six settings of Figure 1.
>
> From the three tables, we can see the Robust Zooming algorithm performs robustly and is significantly better than the original Zooming algorithm under corruption. Compared with RMEL and BoB Robust Zooming algorithms (Figure 1 or Table 3 in Appendix), we can see that the Robust Zooming algorithm with known C is slightly better than the BoB Robust Zooming algorithm in general. And the RMEL is still the best one overall. We believe it is because for Lipschitz bandits elimination-based method turns out to be more powerful in practice.
>
> Thank you very much for your valuable comments on our work. Please do not hesitate to let us know if you have any other questions or concerns of our work.

---

> > ### Author Response · Authors · 2023-08-14
> > **Table 7-11**
> >
> > For your convenience, in addition to the PDF in the above global Rebuttal, we also put Table 7-11 corresponding to our responses to your insightful questions in this Comment section:
> >
> > **Table 7**: Oracle attack (Plot 3 in Figure 1)
> >
> > | Algorithm          | Budget (C) | 60000   | 80000    | 100000   | 120000   | 140000   |
> > |--------------------|------------|---------|----------|----------|----------|----------|
> > | Zooming            | 0          | 3248.54 | 3616.95  | 3705.41  | 3765.19  | 3800.08  |
> > | Zooming            | 3000       | 8730.73 | 11100.13 | 13355.59 | 15374.61 | 17390.93 |
> > | Zooming            | 4500       | 9496.83 | 11893.03 | 14287.24 | 16622.42 | 19015.25 |
> > | RMEL               | 0          | 2589.32 | 2741.95  | 2874.10  | 2941.18  | 2977.19  |
> > | RMEL               | 3000       | 5660.10 | 6515.12  | 6941.57  | 7135.24  | 7205.52  |
> > | RMEL               | 4500       | 6265.09 | 7135.25  | 7564.51  | 7756.00  | 7800.41  |
> > | BoB Robust Zooming | 0          | 3831.94 | 4135.14  | 4314.68  | 4399.01  | 4430.20  |
> > | BoB Robust Zooming | 3000       | 6310.29 | 7035.70  | 7413.45  | 7603.51  | 7656.35  |
> > | BoB Robust Zooming | 4500       | 6932.09 | 7864.41  | 8216.56  | 8401.66  | 8444.31  |
> >
> > **Table 8**: Garcelon attack (Plot 6 in Figure 1)
> >
> > | Algorithm          | Budget (C) | 60000   | 80000    | 100000   | 120000   | 140000   |
> > |--------------------|------------|---------|----------|----------|----------|----------|
> > | Zooming            | 0          | 3248.54  | 3616.95  | 3705.41  | 3765.19  | 3800.08  |
> > | Zooming            | 3000       | 8149.79  | 10199.39 | 11285.41 | 13459.72 | 15230.06 |
> > | Zooming            | 4500       | 13672.00 | 18086.69 | 19441.02 | 22555.63 | 26106.42 |
> > | RMEL               | 0          | 2589.32  | 2741.95  | 2874.10  | 2941.18  | 2977.19  |
> > | RMEL               | 3000       | 2590.77  | 2738.01  | 2879.56  | 2955.07  | 2979.75  |
> > | RMEL               | 4500       | 2872.64  | 3067.71  | 3121.51  | 3193.85  | 3224.00  |
> > | BoB Robust Zooming | 0          | 3831.94  | 4135.14  | 4314.68  | 4399.01  | 4430.20  |
> > | BoB Robust Zooming | 3000       | 4217.74  | 4515.63  | 4684.21  | 4751.72  | 4798.32  |
> > | BoB Robust Zooming | 4500       | 4380.19  | 4645.22  | 4800.05  | 4864.16  | 4901.48  |
> >
> >
> > **Table 9**: Triangle Mean function:
> >
> > | Algorithm      | Budget (C) | Oracle   | Garcelon |
> > |----------------|------------|----------|----------|
> > | Zooming        | 0          | 366.58   | 366.58   |
> > | Zooming        | 3000       | 10883.51 | 10660.17 |
> > | Zooming        | 4500       | 11154.78 | 11487.59 |
> > | Robust Zooming | 0          | 366.58   | 366.58   |
> > | Robust Zooming | 3000       | 510.59   | 509.00   |
> > | Robust Zooming | 4500       | 1211.09  | 749.43   |
> >
> > **Table 10**: Sine Mean function:
> >
> > | Algorithm      | Budget (C) | Oracle  | Garcelon |
> > |----------------|------------|---------|----------|
> > | Zooming        | 0          | 315.94  | 315.94   |
> > | Zooming        | 3000       | 5289.65 | 3174.26  |
> > | Zooming        | 4500       | 5720.30 | 3174.29  |
> > | Robust Zooming | 0          | 315.94  | 315.94   |
> > | Robust Zooming | 3000       | 539.21  | 822.47   |
> > | Robust Zooming | 4500       | 1662.58 | 1048.10  |
> >
> >
> > **Table 11**: Two dim Mean function:
> > | Algorithm      | Budget (C) | Oracle  | Garcelon |
> > |----------------|------------|---------|----------|
> > | Zooming        | 0          | 3248.54 | 3248.54  |
> > | Zooming        | 3000       | 8730.73 | 8149.79  |
> > | Zooming        | 4500       | 9496.83 | 13672.00 |
> > | Robust Zooming | 0          | 3248.54 | 3248.54  |
> > | Robust Zooming | 3000       | 5495.21 | 3770.04  |
> > | Robust Zooming | 4500       | 6439.43 | 3999.21  |

---

> > > ### Comment · Reviewer_4cuU · 2023-08-15
> > > **Response to Rebuttal**
> > >
> > > Thanks for your explanation and experimental results. I don't have any further questions.

---

> > > > ### Author Response · Authors · 2023-08-15
> > > > **Thank you for your response**
> > > >
> > > > We are deeply grateful for your thorough review and insightful comments on our research. And we are delighted to engage in any further discussion with you to address any concern and improve your outlook on our work.

---

### Official Review · Reviewer_Vfr6 · 2023-07-06

**Soundness:** 3 good
**Presentation:** 3 good
**Contribution:** 3 good
**Rating:** 6
**Confidence:** 3

**Summary:**

The article focuses on the problem of Lipschitz bandits in the presence of adversarial corruptions, where an adaptive adversary corrupts the stochastic rewards up to a total budget $C$. Both weak and strong adversaries are considered, where the weak adversary is unaware of the current action before the attack, while the strong one can observe it. This article presents the first line of robust Lipschitz bandit algorithms that can achieve sub-linear regret under both types of adversary, even when the total budget of corruption $C$ is unrevealed to the agent.

**Strengths:**

This work studies Lipschitz bandits in the presence of adversarial corruptions, which are naturally motivated by a broad range of applications with corrupted data. The innovation of the proposed algorithm is that they combined a couple of existing techiniques to design the algorithm,  which help a lot to improve the performance. For example, the authors adopted a removal procedure with multi layers, which could adaptively remove regions at different rates such that the algorithm can torelate corruptions in different levels. This techinique was first proposed in [29] to improve the efficiency of the algorithm for multi-armed bandit in the presense of corruption. Enough intuitions are provided through the presentation of the algorthms and I can easily follow the ideas.

The authors show the advantages of the proposals over past algroithms both theoretically and numerically. The analysis and proofs look correct and intuitive to me.

**Weaknesses:**

1. In additon to the overall budge constraint $C$, the authors require $c_t(x)$ to be bounded by 1. It is stange to put it in Section 3 as an assumption, since the authors claim it is can be easily addressed. Moreover, $c_t(x) > 1$ does not mean the adversary always has enough power to make a suboptimal arm optimal, so I feel it is not straightforward to ignore the case $c_t(x) > 1$.

2. The poposed algorithms (at least some important components) rely on existing techiniques, that somehow reduces the novelty of this work.

**Questions:**

None.

**Limitations:**

Yes.

---

> ### Author Rebuttal · Authors · 2023-08-08
>
> Thank you for your valuable comments on our work. We are pleased to know you think our work is solid and our presentation is clear and intuitive. Please see our response to your concerns as follows:
>
> **Weakness 1: $c_t(x) > 1$?**
>
> The condition $|c_t(x)| < 1$  is used for the proof of Lemma. A.7 in Appendix. We can further relax it by $|c_t(x)| < c$ for any constant $c > 0$ and the same regret bound order in our Theorem 5.1 will still hold. So for simplicity, we just take $c=1$.
>
> Moreover, since we assume that the underlying function $\mu(\cdot)$ is a 1-Lipschitz function and the diameter of space $X$ is no more than 1 (these are common and default assumptions in Lipschitz bandits, e.g. [23]), we know $\Delta(x) \leq 1$ for any $x \in X$. And this fact implies that with $|c_t(x)| = 1$ we could make any arm $x \in X$ optimal or worst in expectation.
> In addition, this condition is also used in robust multi-armed bandit literature [16, 29] against adversarial attacks. Specifically, they make a stronger assumption that the corrupted and stochastic rewards always lie in [0,1] at each round, which naturally implies that the attack value $c_t(x)$ can not exceed 1 in magnitude.
>
> Thank you very much for your careful review. We will remark it in the revision.
>
> **Weakness 2: The poposed algorithms (at least some important components) rely on existing techiniques:**
>
> To the best of our knowledge, our work is the first to study the robust Lipschitz bandits in the presence of adversarial corruption. We acknowledge that our method relies on some existing techniques, e.g. as you mentioned RMEL borrows the multi-layers elimination idea from [29], but we’d also like to highlight that the extension of our paper is highly non-trivial. As we mentioned in lines 203-207, RMEL has to deal with three potential sources of error, and the error from approximation bias is new and rooted in the difficult nature of Lipschitz bandits. Moreover, we developed lower bounds under two types of adversaries in our work, which laid a solid foundation for the future development of this field.
>
>
> We really appreciate your valuable insights and comments on our work, and we are more than happy to engage in any further discussion with you.

---

> > ### Comment · Reviewer_Vfr6 · 2023-08-16
> > **Thanks for your responses**
> >
> > I do not have further questions. Though I will not increase my current rating based on the assessment of the technical novelty and contributions of this work.

---

> > > ### Author Response · Authors · 2023-08-16
> > > **Thank you very much for your review**
> > >
> > > We sincerely appreciate your careful review and value your insightful feedbacks.

---

### Official Review · Reviewer_ecqm · 2023-07-06

**Soundness:** 3 good
**Presentation:** 3 good
**Contribution:** 3 good
**Rating:** 7
**Confidence:** 5

**Summary:**

The paper considers a model of Lipschitz bandits with adversarial corruptions. Two types of adversaries are analysed hand-in-hand: weak adversary which may not have knowledge of the current action of the learner before injecting its corruption into the reward that is actually observed by the learner, and strong adversary which has knowledge of the current action of the learner and is hence more challenging to defend against. Irrespective of its nature, the adversary may only inject a total corruption of $C$ over $T$ rounds of sampling. The learner may or may not have prior knowledge of the value of $C$, and is tasked with the goal of minimizing the cumulative regret over the horizon $T$.

The paper analyzes the following scenarios.
1. The learner has prior knowledge of $C$ (Section 4).
2. The learner has no prior knowledge of $C$, and must deal with a weak adversary (Section 5.1).
3. The learner has no prior knowledge of $C$, and must deal with a strong adversary (Section 5.2).

For each of the above scenarios, the paper presents a lower bound on the cumulative regret and one or more algorithms for cumulative regret minimization. For scenario 1, the paper proposes a robust zooming algorithm, inspired by the UCB algorithm, albeit for a continuum of arms. For scenario 2, a robust multi-layer elimination algorithm is proposed. For scenario 3, three different algorithms are proposed. The paper derives upper bounds on the cumulative regret for each of the algorithms under every scenario. A noteworthy aspect of the lower and the upper bounds are their dependence on the corruption budget $C$, regardless of whether $C$ is known or unknown.

**Strengths:**

1. The paper extends the analysis of prior works on adversarial corruptions in stochastic multi-armed bandits to handle a continuum of arms; most prior works in bandits deal with only finitely many arms.
2. The authors' presentation of the key results of the paper in Table 1, as a quick, one-stop summary of all the important results of the paper, is commendable. A mere cursory glance of Table 1 suffices for an expert reader to recall the important contributions of the paper.
3. The authors' effort towards extracting the key ideas from several prior works (notably [23], [29], [31], and [11]) and applying these ideas to solve the problem at hand (with a continuum of arms) is truly commendable.
4. The dependence of the lower and the upper bounds on the corruption budget $C$, the zooming dimension $d_z$, and the covering dimension $d$ is noteworthy.

**Weaknesses:**

The section on Preliminaries (Section 3) needs to be supplemented with additional details and rewritten to correct some glaring errors. Below are some points I have mentioned based on my detailed reading of the paper.

1. In the definitions of the covering dimension and the zooming dimension, $\exists c>0$ should be replaced with $\exists \alpha>0$.
2. For any $r \in (0,1]$, the "$r$-covering" of a compact set must be defined formally, noting that it appears frequently in Algorithm 2 (where phrases such as "$1/2$-covering", "$1/2^{m_l+1}$-covering" are used).
3. It is better to denote the covering dimension by $d_c$ instead of $d$; the latter could be referred to as the Euclidean dimension, following the lines of [23]. This avoids strange (yet correct) phrases such as "$d=d$" appearing on line 136, which as per the suggested notations would read "$d_c=d$".
4. It is not quite clear why $d_z \leq d$. The definition of the covering dimension $d$ entails using balls of radius at most $r$ for the covering, whereas that of the zooming dimension $d_z$ entails using balls of radius at most $r/16$ for the covering. The authors must clarify why $d_z\leq d$, as this is crucial for comparing the upper bound of Theorem 5.3 with the lower bound of Theorem 4.2.
5. In continuation to the previous point, the paper [23] alludes to the zooming dimension as "the smallest $q$ such that for every $r>0$, only $O(r^{-q})$ sets of radius $r/16$ are required to cover the set $\lbrace x \in \mathcal{X}: r\leq \Delta(x) \leq 2r \rbrace$ (this is rephrased in terms of the notations used in the current paper); see, for instance, the last paragraph in [23, pp. 30:5]. In contrast, the authors define the zooming dimension as "the smallest $q$ such that for every $r>0$, only $O(r^{-q})$ sets of radius $r/16$ are required to cover the "$r$-optimal set" $\lbrace x \in \mathcal{X}: \Delta(x) \leq r \rbrace$", which differs from the one in [23]. Can the authors elaborate on why they chose to define the zooming dimension differently from [23], and what are the implications of this difference on their results?
6. I am not very convinced about the nature of adversarial corruptions used for the simulations. I cannot readily see if "Oracle" and "Garcelon" attacks respect the adversarial attack model $\max_{x \in \mathcal{X}} |c_t(x)| \leq 1$ for all $x,t$ that forms the heart of the paper. In fact, in Garcelon, using a Gaussian random variable as the corrupted reward has the implication that the corruption is unbounded, thereby not adhering to the proposed adversarial attack model. In my opinion, it is quite difficult to judge in a fair manner the performance of the proposed algorithms on adversarial attack models that differ from the underlying model proposed in Section 3.
7. Continuing on the above point, why do the authors not consider in their simulations an explicit corruption function that satisfies the requirement $\max_{x \in \mathcal{X}} |c_t(x)| \leq 1$ for all $x,t$? For instance, say $c_t(x)=\sin(2\pi xt/T)$, $x \in [0,1]$, $t \in \{1, \ldots, T\}$. Is incorporating such functions into the simulations challenging? If so, what exactly are the challenges? Simulating an attack model that matches exactly with the description of Section 3 would significantly enhance the purpose of simulations, lead to a fair evaluation of the performances of the proposed algorithms, and demonstrate the strength/utility of the proposed algorithms.

**Questions:**

1. The authors provide separate algorithms for the case when the adversary is weak and when the adversary is strong. While I appreciate the effort put into analysing both of these scenarios separately in the paper, I have the following question: in practice, is the algorithm (more specifically, the learner) aware of the nature of the adversary beforehand? Are there practical scenarios where the learner has prior knowledge about whether the adversary is weak/strong? I would imagine that it is very unlikely that such prior knowledge may actually be available in practice, in which case it would be reasonable for the learner to simply design an algorithm that works against the strong adversary. Can the authors comment about this (if possible also include a sentence or two on this point in the paper)?
2. The paper [23] alludes to the zooming dimension as "the smallest $q$ such that for every $r>0$, only $O(r^{-q})$ sets of radius $r/16$ are required to cover the set $\lbrace x \in \mathcal{X}: r\leq \Delta(x) \leq 2r \rbrace$ (this is rephrased in terms of the notations used in the current paper); see, for instance, the last paragraph in [23, pp. 30:5]. In contrast, the authors define the zooming dimension as "the smallest $q$ such that for every $r>0$, only $O(r^{-q})$ sets of radius $r/16$ are required to cover the "$r$-optimal set" $\lbrace x \in \mathcal{X}: \Delta(x) \leq r \rbrace$", which differs from the one in [23].
Can the authors elaborate on why they chose to define the zooming dimension differently from [23], and what are the implications of this difference on their results?
3. This is more of an observation than a question. The upper bound on cumulative regret derived in Theorem 5.1 relies on lower bounding the probability of the event $\Phi$ as defined in the proof sketch. In the definition of $\Phi$, the layers whose tolerance values exceed the unknown corruption budget $C$ are considered. To me, this seems like an "artificial" way of introducing $C$ into $\Phi$ in order to derive an upper bound that depends on $C$. However, this also suggests that without bringing $C$ into the definition of $\Phi$, an upper bound on cumulative regret that does not depend on $C$ may be derived. Such an upper bound may be relevant to problems where $C$ is potentially infinite (i.e., the adversary may corrupt the rewards in an unbounded fashion), thereby broadening the scope of the current work (when $C=+\infty$, the upper bounds in the paper become degenerate). However, I envisage that the analysis of the case $C=+\infty$ could be hard. Perhaps, this could be a direction of future work.

**Limitations:**

See points 6,7 under "weaknesses".

---

> ### Author Rebuttal · Authors · 2023-08-08
>
> Thank you for your valuable comments on our work. We are happy to know you think our contributions are commendable and our presentation is clear. Please see our response to your concerns as follows:
>
> **Weakness 1-3: Typos and notations:**
>
> We are grateful for your meticulous review. We will correct the typos and clarify our notations in the revision.
>
> **Weakness 4: Why $d_z \leq d$?**
>
> For recall, we write relevant notations here in a metric space $(X,D)$:
>
> r-zooming number $N_z(r)$ is the minimal number of balls of radius not more than $r/16$ required to cover the $r$-optimal region defined as {$ x\in X:\Delta(x) \leq r $}. r-covering number $N_c(r)$ is the minimal number of balls with radius of no more than $r$ required to cover $X$. Zooming dimension: $d_z = \min$ { $q \geq 0: \exists \alpha_1>0, N_z(r) \leq \alpha_1 r^{-q}, \forall r \in (0,1]$}. Covering dimension: $d_c = \min$ \{ $q \geq 0: \exists \alpha_2>0, N_c(r) \leq \alpha_2 r^{-q}, \forall r \in (0,1]$}.
>
> First, we know $N_z(16r)$ is no larger than $N_c(r)$ for arbitrary $r > 0$ since {$x\in X:\Delta(x) \leq 16r$} is a subset of $X$. Therefore, by the definition of the covering dimension $d_c$, there exists some $\alpha_2 >0$ s.t. $N_z(16r) \leq \alpha_2 r^{-d_c}, \forall r > 0.$ This is identical to $N_z(16r) \leq \alpha_2 16^{d_c} (16r)^{-d_c}, \forall r > 0,$ which means that
> $$N_z(r) \leq \alpha_2 16^{d_c} (r)^{-d_c} = \alpha_1 (r)^{-d_c}, \forall r > 0, \alpha_1 = \alpha_2 16^{d_c}.$$
> Note $16^{d_c}$ is also a constant free of $r$, and based on the definition of the zooming dimension $d_z$, it holds that $d_z$ is at most $d_c$.
>
> In conclusion, the zooming dimension will be the same using balls of radius at most $cr$ for the covering for arbitrary $c>0$ of constant scale. This fact is commonly used in the Lipschitz bandit literature and is directly illustrated in [23] (right after Theorem 1.3), where different radii for covering number and zooming number are used.
>
> Thank you for your insightful question, and we will put this detailed explanation in our revision.
>
> **Weakness 5, Question 2: Different notations about the Zooming dimension:**
>
> First, we explain why we use the set {$x \in X: \Delta(x) \leq r$} in our paper: we use this notation for a clear and easy-to-follow proof. Specifically, based on Lemma A.12 in Appendix, we can show that any arm played after the epoch $m$ would at most incur a regret of $16/2^m$, and hence we could bound the number of active regions at epoch $m$ by the order $O(2^{d_z m})$ with the definition. And then the total regret incurred at that epoch could be bounded.
>
> Second, we can show that the zooming dimension under these two different sets is equivalent. To keep consistent with the notations in our paper, we further denote the r-zooming number used in [23] as $N_z’(r)$ for the set {$x \in X: r/2 < \Delta(x) \leq r$}, and the corresponding zooming dimension for $N_z’(r)$ as $d_z’$. Naturally, it holds that $d_z’ \leq d_z$ since {$x \in X: r/2 < \Delta(x) \leq r$}$ \subset$ {$x \in X: \Delta(x) \leq r$}, and hence it suffices to show that $d_z’ \geq d_z$ holds as well.
> If $d_z = 0$ it naturally holds that $d_z’ = 0 = d_z$. If $d_z > 0$, note by definition there exists $\alpha_1, \alpha_2 >0$ s.t. $N_z(r) \leq \alpha_1 r^{-d_z}, N_z’(r) \leq \alpha_2 r^{-d_z’}$. Since {$x \in X: \Delta(x) \leq r$}$ = ${$x \in X:  r/2 < \Delta(x) \leq r$}$ \bigcup ${$x \in X: \Delta(x) \leq r/2$}, and the coverings of A and the coverings of B will cover A $\cup$ B, we have that $N_z(r) \leq N_z’(r) + N_z(r/2), \forall r \in (0,1]$. This implies that
> $$(1/r)^{d_z’-d_z} \geq (\alpha_1/\alpha_2) \times (1-0.5^{d_z})), \forall r \in (0,1], d_z > 0.$$
> And hence we have $d_z’ \geq d_z$. Conclusively, it holds that $d_z’ = d_z$. Note our definition of zooming dimension is the same as “near-optimality dimension” introduced in another most-cited Lipschitz bandit paper [8], where the authors also mention the equivalence between the zooming dimension used in [23] and the near-optimality dimension from section 5.2 and the proposed lower bound. We will include a remark on this topic in the revision. Thank you very much for helping improve our paper.
>
> [8] X-armed Bandits. S. Bubeck et al., JMLR 2011.
>
> **Weakness 6: Experiments are not consistent with $|c_t(x)| \leq 1$:**
>
> Thank you very much for your suggestion. To be consistent with our theory, we modified both attacks in our simulations, and restricted that the attack volume to be at most one (and if the attack volume was greater than 1, we truncated it into 1). For example, if the stochastic reward is 0.8, and the Garcelon generated a corrupted reward of -0.5, then we will use 0.8-1=-0.2 as the observed reward after corruption.
>
> We re-ran all six simulations in Figure 1 and then output the final cumulative regrets under both the original attacks used in our paper and the modified attacks in Table 1-3 in the PDF in the above Global Rebuttal. From the tables, we can observe that algorithms behave similarly under truncated Oracle and Garcelon attacks and hence we can reach the same conclusion as in our paper in lines 335-344: Zooming becomes futile while our proposed algorithms, especially RMEL, are still robust to the contaminated environment.
>
> **Due to the space limit and a single allowed Rebuttal, we will respond to your other insightful questions by posting new Comments in the very beginning of Discussion phase. We are sincerely grateful for your valuable comments.**

---

> > ### Author Response · Authors · 2023-08-10
> > **Responses to your remaining valuable questions**
> >
> > **Weakness 7: Why not $sin()$ attack?**
> >
> > First, we’d like to explain why we choose Oracle and Garcelon attacks in our paper: This is because these two attacks are well studied in the previous literature with theoretical guarantees for multi-armed bandits and linear bandits, and it has been verified that both attacks could destroy the performance of stochastic bandit algorithms in practice. As the Lipscthiz bandit is an extension of a multi-armed bandit with infinitely many arms in a metric space, we just naturally extend these two attacks into our setting, and we expect these two attacks will also be detrimental to stochastic Lipschitz bandits. From Figure 1 and the additional results in our response to weakness 6 (truncated Oracle and Garcelon attacks), we could also observe the effectiveness of these two types of attacks since the Zooming algorithm becomes futile and achieves linear regret under both attacks. Intuitively, these two attacks are malicious since they contaminate the stochastic rewards generated from “good arms” by pushing them to the very bottom or modifying them into random noise.
> >
> > Moreover, we just implemented the explicit corruption function $c_t(x) = \sin(2 \pi x t/T)$ as the adversarial attack in our experiments under the three underlying functions in Figure 1, and we inject the attack with probability 0.1 at each round. Tables 4-6 in the above PDF in the global Rebuttal show the cumulative regrets of different algorithms under three reward functions. From the tables, we can observe that both our algorithms, especially RMEL, still yield more robust performance under different volumes of corruption compared with Zooming does. We can also observe that this type of attack is not as malicious as Oracle and Garcelon, and we believe it is because the attack volume $c_t(x)$ is very small in the beginning (sin(0)=0), and hence the Zooming algorithm has sufficient time to learn the underlying reward function in the early stages.
> >
> > **Question 1: strong vs weak adversaries in practice:**
> >
> > From the theoretical side, We study two types of adversaries in our paper to be consistent with the existing literature on stochastic bandits under adversarial corruption.
> >
> > In practice, we feel we can have a better understanding of this problem through an analogy with adversarial attacks in deep learning [10]: a white-box attacker has complete knowledge about the target model before the attack, which is like the strong adversary in the bandit knowing which arm the target model is going to pull before the attack. On the other hand, a black-box attacker doesn’t know the target model, which is like the weak adversary in the bandit unaware of the current action from the target bandit algorithm. In practice, a defender can consider defense against either black-box or white-box attacks according to whether the learning algorithm can be revealed to the attacker. Most of the practical systems won't reveal the learning model to the attacker, so in practice one often considers black-box attacks. However, defenses against white-box attacks are also widely studied since they provide a better safety guarantee.
> >
> > Following the above arguments, for the robustness of bandit algorithms, we believe which adversary to use depends on whether the bandit algorithm can be potentially revealed, and how much safety guarantee we want to have. For example, if model developers want to perform robustness testing, then the strong adversary can be used since it can evaluate the worst-case performance of a model. On the contrary, for attackers that are not able to hack into the system and observe which arm our underlying model is going to choose, we can regard them as weak adversaries instead and make corresponding defense.
> >
> > **Question 2: derive an algorithm whose regret bounds are free of $C$**
> >
> > Thank you very much for your insightful idea and careful review on our work. We also believe there is some improvement space regarding our Theorem 5.1 where we artificially construct the set $\Phi$, and we feel a direct and natural extension is to study the case when we don’t have $|c_t(x)|$ is bounded. Since we believe a malicious attacker will try to contaminate as many rounds as possible and hence will not allocate too many budgets in any single round. In other words, we think a clever adversary may bound $|c_t(x)|$ itself. Therefore, it might be “easier” if $|c_t(x)|$ is unbounded. However, our analysis in Theorem 5.1 hinders us from solving this “easier” case.
> >
> >
> > We also believe it is interesting to study the case when C is infinitely large, or $C > T$. And we feel the first natural question is how to define a reasonable metric, like a different type of cumulative regret. Since the original cumulative regret $Regret_T = T \mu(x_*) - \sum_{t=1}^T \mu(x_t)$ may be unbounded when the adversary could contaminate every round. Due to the time limit, we will leave it as an interesting future work.

---

> > ### Author Response · Authors · 2023-08-14
> > **Table 1-6**
> >
> > For your convenience, in addition to the PDF in the global Rebuttal, we also put Table 1-6 w.r.t. our responses to your valuable questions here:
> >
> > **Table 1**: Triangle (Figure 1, 4)
> > | Algorithm          | Budget (C) | Oracle   | Garcelon |  Modified Oracle | Modified Garcelon |
> > |--------------------|------------|----------|----------|------------------|-------------------|
> > | Zooming            | 0          | 366.58   | 366.58   | 366.58           | 366.58            |
> > | Zooming            | 3000       | 10883.51 | 10660.17 | 10824.85         | 10530.29          |
> > | Zooming            | 4500       | 11153.78 | 11487.59 | 11140.29         | 11491.21          |
> > | RMEL               | 0          | 512.46   | 512.46   | 512.46           | 512.46            |
> > | RMEL               | 3000       | 921.95   | 504.78   | 918.29           | 512.10            |
> > | RMEL               | 4500       | 928.27   | 1542.17  | 925.91           | 1540.89           |
> > | BoB Robust Zooming | 0          | 461.16   | 461.16   | 461.16           | 461.16            |
> > | BoB Robust Zooming | 3000       | 495.06   | 531.37   | 499.17           | 526.41            |
> > | BoB Robust Zooming | 4500       | 1323.97  | 736.85   | 1330.02          | 739.61            |
> >
> > **Table 2**: Sine (Figure 2, 5)
> > | Algorithm          | Budget (C) | Oracle   | Garcelon |  Modified Oracle | Modified Garcelon |
> > |--------------------|------------|----------|----------|------------------|-------------------|
> > | Zooming            | 0          | 315.94  | 315.94   | 315.94           | 315.94            |
> > | Zooming            | 3000       | 5289.65 | 3174.26  | 5284.19          | 3169.01           |
> > | Zooming            | 4500       | 5720.30 | 3174.29  | 5710.35          | 3178.96           |
> > | RMEL               | 0          | 289.86  | 289.86   | 289.86           | 289.86            |
> > | RMEL               | 3000       | 442.66  | 289.29   | 438.17           | 294.20            |
> > | RMEL               | 4500       | 862.90  | 332.71   | 855.38           | 335.17            |
> > | BoB Robust Zooming | 0          | 435.44  | 435.44   | 435.44           | 435.44            |
> > | BoB Robust Zooming | 3000       | 414.54  | 746.96   | 410.92           | 748.60            |
> > | BoB Robust Zooming | 4500       | 1887.35 | 1148.09  | 1875.11          | 1158.62           |
> >
> > **Table 3**: Two Dim (Figure 3, 6)
> > | Algorithm          | Budget (C) | Oracle  | Garcelon |  Modified Oracle | Modified Garcelon |
> > |--------------------|------------|---------|----------|------------------|-------------------|
> > | Zooming            | 0          | 3248.54 | 3248.54  | 3248.54          | 3248.54           |
> > | Zooming            | 3000       | 8730.73 | 8149.79  | 8723.18          | 8153.53           |
> > | Zooming            | 4500       | 9496.83 | 13672.00 | 9496.83          | 13672.00          |
> > | RMEL               | 0          | 2589.32 | 2589.32  | 2589.32          | 2589.32           |
> > | RMEL               | 3000       | 5660.10 | 2590.77  | 5665.44          | 2587.09           |
> > | RMEL               | 4500       | 6265.09 | 2872.64  | 6274.96          | 2869.74           |
> > | BoB Robust Zooming | 0          | 3831.94 | 3831.94  | 3831.94          | 3831.94           |
> > | BoB Robust Zooming | 3000       | 6310.29 | 4217.74  | 6312.19          | 4222.60           |
> > | BoB Robust Zooming | 4500       | 6932.09 | 4380.19  | 6928.63          | 4388.97           |
> >
> > **Table 4**: Triangle:
> > | Algorithm          | Budget (C) | Regret |
> > |--------------------|------------|--------|
> > | Zooming            | 0          | 366.58 |
> > | Zooming            | 3000       | 588.02 |
> > | Zooming            | 4500       | 1499.11 |
> > | RMEL               | 0          | 512.46 |
> > | RMEL               | 3000       | 508.94 |
> > | RMEL               | 4500       | 642.17 |
> > | BoB Robust Zooming | 0          | 461.16 |
> > | BoB Robust Zooming | 3000       | 511.14 |
> > | BoB Robust Zooming | 4500       | 642.08 |
> >
> > **Table 5**: Sine:
> > | Algorithm          | Budget (C) | Regret |
> > |--------------------|------------|--------|
> > | Zooming            | 0          | 315.94 |
> > | Zooming            | 3000       | 539.62 |
> > | Zooming            | 4500       | 1072.21 |
> > | RMEL               | 0          | 289.86 |
> > | RMEL               | 3000       | 284.11 |
> > | RMEL               | 4500       | 296.50 |
> > | BoB Robust Zooming | 0          | 435.44 |
> > | BoB Robust Zooming | 3000       | 455.72 |
> > | BoB Robust Zooming | 4500       | 531.13 |
> >
> > **Table 6**: Two dim:
> > | Algorithm          | Budget (C) | Regret  |
> > |--------------------|------------|---------|
> > | Zooming            | 0          | 3248.54 |
> > | Zooming            | 3000       | 4412.59 |
> > | Zooming            | 4500       | 5983.09 |
> > | RMEL               | 0          | 2589.32 |
> > | RMEL               | 3000       | 2594.16 |
> > | RMEL               | 4500       | 2670.15 |
> > | BoB Robust Zooming | 0          | 3831.94 |
> > | BoB Robust Zooming | 3000       | 3900.31 |
> > | BoB Robust Zooming | 4500       | 4198.11 |

---

> > ### Comment · Reviewer_ecqm · 2023-08-15
> > **Response to authors' rebuttal**
> >
> > I thank the authors for their detailed response to my comments and for carrying out additional experiments heeding to my suggestions.
> > Indeed, the new experiments, while conforming to the system model outlined in Section 3, affirm that the algorithms proposed in the paper have superior performance over the classical Zooming algorithm.
> >
> > I thank the authors for providing a justification for why $d_z$ is independent of the "radius" used in the definition of the zooming number.
> > In connection with a proof of this fact, I only have the following minor suggestion for the authors.
> > The authors note that $(1/r)^{d_z^\prime-d_z} \geq (\alpha_1/\alpha_2) \times (1-0.5^{d_z}), \forall r \in (0,1], d_z > 0$, and use this to conclude $d_z^\prime \geq d_z$. However, I do not readily see how their conclusion follows from the preceding relation.
> >
> > In my opinion, a more straightforward way to arrive at the authors' conclusion is to note that
> > $$
> > N_z(r) \leq N_z’(r) + N_z(r/2) \leq \alpha_1 r^{-d_z^\prime} + \alpha_2 (r/2)^{-d_z} \leq  (\alpha_1+ \alpha_2 \\, 2^{d_z}) r^{-d_z^\prime} \quad \forall r >0,
> > $$
> > where the last inequality follows because $d_z \geq d_z^\prime$ trivially. From the inequality above, noting that $\alpha_1 + \alpha_2 \\, 2^{d_z}$ is independent of $r$, we get $d_z \leq d_z^\prime$.
> >
> > ---
> >
> > In light of the authors' detailed rebuttal and the results of the new experiments, I am willing to increase my score from 6 to 7.

---

> > > ### Author Response · Authors · 2023-08-15
> > > **Thank you for your careful review and valuable suggestion**
> > >
> > > We sincerely value the time you've dedicated to reviewing our work and deeply appreciate your thoughtful suggestions, which have significantly enhanced the quality of our paper.

---

### Official Review · Reviewer_Mz32 · 2023-07-06

**Soundness:** 3 good
**Presentation:** 3 good
**Contribution:** 3 good
**Rating:** 5
**Confidence:** 3

**Summary:**

This paper studies Lipschitz bandits with adversarial corruptions, where the reward of the pulled arm can be maliciously corrupted by an adversary. The authors consider both weak adversary and strong adversary and present robust algorithms for each setting. Under the weak adversary setting, the paper proposes an elimination-based algorithm and derives a regret bound with nearly optimal dependence on the time horizon. For strong adversary, the paper shows that a simple modification of the Zooming algorithm can attain sub-linear regret when the corruption budget is known in advance and applies several model selection methods to handle unknown corruption budget. The paper also provide numerical results to demonstrate the advantage of the proposed algorithm in corrupted scenarios.

**Strengths:**

The paper is the first to study Lipschitz bandits with adversarial corruptions and proposes several algorithms with both sub-linear regret bounds and promising empirical performance.

**Weaknesses:**

1. For weak adversary, the regret bound of the proposed algorithm has a multiplicative dependence on the corruption budget, which is unsatisfactory and an additive dependence is desired.

2. For strong adversary, the proposed algorithm is mainly a combination of existing methods with little novelty. The derived regret bounds are also weak as they depend on the covering dimension rather than the zooming dimension.

**Questions:**

In the experiment with two dim mean function under oracle attack, why the performance of RMEL is better than Zooming?

**Limitations:**

The authors have discussed the limitations of the proposed algorithms.

---

> ### Author Rebuttal · Authors · 2023-08-08
>
> Thank you for your valuable questions, and please see our responses to your concerns below:
>
> **Weakness 1: For weak adversaries, the regret bound of the proposed algorithm has a multiplicative dependence on C, an additive dependence is desired.**
>
> We also state this point as a limitation of our paper in lines 243-245. And we believe it is very challenging to introduce an algorithm whose regret bound has an additive dependence on the corruption budget C under the Lipschitz bandits. For the stochastic linear bandit whose regret is $d  \sqrt{T}$, we can control its regret by $d \sqrt{T} + dC$ under corrupted value C. However, unlike the linear bandit, the regret of stochastic Lipschitz bandit is $T^{(d_z+1)/(d_z+2)}$, which scales exponentially with the zooming dimension under the base $T$. Therefore, it is naturally very challenging to separate the zooming dimension $d_z$ and $T$ into two terms for the regret analysis of the corrupted Lipschitz bandit. Therefore, whether it is possible to propose an algorithm that has an additive dependence on $C$ in a separate term free of $T$ is still unclear.
>
> We’d like to re-emphasize that the regret analysis of our RMEL algorithm is highly non-trivial. As we mentioned in lines 203-207, it is difficult to apply the elimination idea on the Lipschitz bandit under corruption since there are potentially three sources of error. Besides, both the zooming dimension $d_z$ and corruption $C$ are unrevealed to us, but our RMEL could adaptively attain the regret bound depending on these two terms, and could achieve the optimal regret bound in the uncorrupted case up to some logarithmic terms (line 241).
>
> **Weakness 2: For strong adversary, the proposed algorithm is mainly a combination of existing algorithms. Regret depends on the covering dimension rather than the zooming dimension.**
>
> We present a line of algorithms under the strong adversary and unknown budget $C$ for the completeness of our work.
>
> For the strong adversary, we first studied the case when the corruption budget $C$ is known, and propose an algorithm whose regret bound matches the lower bound we deduce in Theorem 4.2, which implies that our proposed algorithm is minimax optimal. For the unknown corruption budget case, we developed the following lower bound under the suggestion from Reviewer DQs2:
>
> Theorem: For any algorithm, when there is no corruption, we denote $R^0_T$ as the upper bound of cumulative regret in T rounds under our problem setting described in Section 3, i.e. $Regret_T \leq R^0_T$ with high probability, and it holds that $R^0_T = o(T)$. Then under the strong adversary and unknown attacking budget $C$, there exists a problem instance on which this algorithm will incur linear regret $\Omega(T)$ with probability at least $0.5$, if $C = \Omega(R^0_T/4^{d_z})  = \Omega(R^0_T)$.
>
> And we will put it and its proof in the next revision. The main contribution of our work is to propose lower bounds of different cases and the RMEL algorithm, which works efficiently in theory and in practice. For completeness, we introduce two algorithms under the strong adversary and unknown budget setting. Introducing algorithms under this setting with better regret bound will be a challenging future direction.
>
> **Question: In experiment with two dim mean function under oracle attack, why RMEL is better than Zooming?**
>
> We are not sure if we fully understand your question (we apologize for that), and hence we offer two answers from different understandings:
>
> Why RMEL is better than Zooming under corruptions:
>
> The Zooming algorithm is designed for the purely stochastic Lipschitz bandit setting, and will become futile in both theory and practice if the stochastic rewards are contaminated by adversaries. This can be observed by all the experimental results in Figure 1, where the Zooming algorithm suffers from linear cumulative regret under a mild volume of attacks. On the contrary, our RMEL algorithm is designed to defend against the corruption from adversaries in the Lipschitz bandit problem, and can achieve a decent sub-linear regret bound that is adaptive to the attacking budget $C$. From Figure 1, we can see that RMEL yields very robust results under various scenarios with different volumes of attacks.
>
> Why RMEL is better than Zooming without corruptions:
>
> Note in theory, our RMEL could attain the same optimal regret bound as Zooming up to some logarithmic terms when there is no corruption as we mention in lines 240-242 in our paper. In other words, the regret bound of our RMEL is adaptive and no worse than that of the Zooming algorithm under no corruption up to some log terms. Therefore, we can expect RMEL and Zooming to perform similarly when there is no attack, which also coincides with our experimental results: Under the non-corrupted case, in two settings (triangle, sine) Zooming is better, while in one setting (two dim) RMEL outperforms Zooming.
>
> Thank you very much for your valuable insights. Please let us know if our response has decently resolved your concern and improved your opinion of our work. And we are more than happy to take any additional questions from you.

---

> > ### Comment · Reviewer_Mz32 · 2023-08-18
> >
> > Thanks for the response. It addresses my concerns and I will increase the score accordingly.

---

> > > ### Author Response · Authors · 2023-08-18
> > > **Thank you for your careful review**
> > >
> > > We sincerely appreciate your comprehensive feedback and valuable insights on our study. We are delighted to engage in further discussions with you to address any concerns and enhance your perspective on our work.

---

### Official Review · Reviewer_DQs2 · 2023-07-06

**Soundness:** 4 excellent
**Presentation:** 4 excellent
**Contribution:** 4 excellent
**Rating:** 7
**Confidence:** 5

**Summary:**

The paper studied robust Lipschitz bandits under two types of corruptions - a weak adversary who perturbs the reward function before observing the selected action; and a strong adversary who perturbs the instantiated reward of the selected action. When the attack budget is known, the paper proposed to use enlarged confidence budget, and the regret upper bound of the corresponding algorithms attains the lower bound derived in the paper. When the attack budget is unknown, the paper developed two robust algorithms targeting the weak and strong adversary separately. For the weak adversary, both upper and lower regret bound are derived. There is a gap between these two bounds. For the strong adversary, the paper derived upper bound of the regret. Experiments on synthetic datasets demonstrate the effectiveness of the proposed robust algorithms.

**Strengths:**

(1). The topic of studying robustness in bandits is a timely topic. This extends prior works on robust bandit to the Lipschitz bandit scenario, which pushes the frontier in this domain.

(2). The work is relatively complete - two types of adversaries are considered, and the attack budget are defined accordingly; two threat models are considered, including known and unknown attack budget. Besides that, both upper and lower regret bounds are derived, which shows that the robust algorithm designed in this paper is close to optimal.

**Weaknesses:**

I don't see major weaknesses with the paper. A minor weakness is that for the strong adversary and the unknown attack budget setting, the paper is missing a regret lower bound.

**Questions:**

Can the authors provide some discussions on whether it's possible to derive regret lower bound for the strong adversary and unknown attack budget setting?

**Limitations:**

Yes

---

> ### Author Rebuttal · Authors · 2023-08-08
>
> Thank you for your insightful comments on our work. We are happy to know you find our work pushes the frontier on a timely topic and is relatively complete. Please see our response to your question as follows:
>
> **Can the authors provide some discussions on whether it's possible to derive regret lower bound for the strong adversary and unknown attack budget setting?**
>
> Thank you for your constructive question. Inspired by Theorem 4 in [1], which claims the regret lower bound of linear bandits in the two-dimensional space with two arms (d=2,k=2) under strong adversary and unknown budget C, in the past few days we successfully extended their result to the continuum-armed Lipschitz bandit setting and deduced the following Theorem:
>
> Theorem: For any algorithm, when there is no corruption, we denote $R^0_T$ as the upper bound of cumulative regret in T rounds under our problem setting described in Section 3, i.e. $Regret_T \leq R^0_T$ with high probability, and it holds that $R^0_T = o(T)$. Then under the strong adversary and unknown attacking budget $C$, there exists a problem instance on which this algorithm will incur linear regret $\Omega(T)$ with probability at least $0.5$, if $C = \Omega(R^0_T/4^{d_z})  = \Omega(R^0_T)$.
>
> Proof sketch: we use a similar proof structure as in the proof of our Theorem 4.2 and Theorem 5.2. Specifically, we first study the case $d_z = 0$, and use the problem setting $A_1$ where the Lipschitz function $f(x) = 0.25-|x-0.25|$ if $x \in [0,0.5]$ and $f(x) = 0$ otherwise on the space ([0,1], ||), and we assume there is no random noise. For any algorithm with $Regret_T \leq R^0_T$, with Markov inequality we know that $P(\text{the number of pulls in }(0.5,1] \leq 8 R^0_T) \geq 0.5$. We introduce a new problem instance $A_2$: $g(x) = 0.25-|x-0.25|$ if $x \in [0,0.5]$ and $g(x) = 0.5-|x-1|$ if $x \in (0.5,1]$, and we have a strong adversary to attack as follows: whenever the arm in $(0.5,1]$ is pulled, the adversary changes the reward to 0 before the total corruption $C= 4 R^0_T = \Omega(R^0_T)$ is used up. Then the agent can not tell the difference between $A_1$ and $A_2$ until the number of times to select arms in $(0.5,1]$ reaches at least $C/0.5=8 R^0_T$. Therefore, with probability at least $0.5$, the cumulative regret is at least $0.25(T - 8 R^0_T) = \Omega(T)$.
>
> For the case $d_z > 0$, we can similarly construct the instance in the space $([0,1]^{d=\lceil 2 d_z \rceil}, ||\cdot||)$ where the norm $||\cdot|| = ||\cdot|| _{\infty}$. And the function $f_1(x) = 4^{-\frac{d}{d-d_z}} - ||x-(0.25,\dots,0.25)||^{\frac{d}{d-d_z}}, x_i \in [0,0.5]$. And then we could use the Pigeonhole principle to argue that with probability at least $1/2$ the number of times to pull arms in some region can be upper bounded by $O(R^0_T/2^d)$. By constructing a new instance where the reward function takes maximum value in this region, we can prove our Theorem using a similar argument as above.
>
>
> We know the optimal worst-case regret in the stochastic Lipschitz bandit problem is of order $R^0_T=O (\ln(T)^{(1)/(d_z+2)}T^{(d_z+1)/(d_z+2)}) = \tilde O (T^{(d_z+1)/(d_z+2)}) $, and hence this Theorem suggests that for any algorithm (e.g. Zooming algorithm) that could attain this regret bound under the non-corruption setting, it must suffer linear regret if unknown corruption is of order $\Omega(\ln(T)^{(1)/(d_z+2)}T^{(d_z+1)/(d_z+2)})$ and comes from the strong adversary.
>
> We will put this Theorem and its detailed proof with explanations in the revision. Thank you very much for helping improve our work.
>
> [1] Stochastic Linear Bandits Robust to Adversarial Attacks, I Bogunovic et al., AISTATS 2021.

---

### Author Rebuttal · Authors · 2023-08-09

We sincerely appreciate the time and effort you dedicated to evaluating our manuscript, and we value the insights provided to enhance the quality of our work. We are happy to know that you find our work studies an essential and timely topic, with solid theory analysis and promising performance in practice

The Attached PDF contains the tables used in our Rebuttal.

Specifically,

Tables 1-3 are for Reviewer ecqm (Weakness #6), where we reran our experiments in Figure 1 by restricting that the corruption values from both attacks are at most $1$ at each round.

Table 4-6 are for Reviewer ecqm (Weakness #7), where we implement the explicit corruptions function $c_t(x) = \sin(2 \pi x t/T)$ as the adversarial attack in our experiments under the three underlying functions in Figure 1.

Table 7-9 are for Reviewer 4cuU (Weakness #2), where we show the convergence of regrets of our algorithms in the long run under the two dim mean reward functions.

Table 10-11 for  Reviewer 4cuU (Question #3), where we show the promising performance of Robust Zooming algorithm given the value of $C$ is revealed to the agent.

---

### Decision · Program_Chairs · 2023-09-21

**Decision:**

Accept (poster)

**Comment:**

The paper advances the field by extending the analysis of adversarial corruptions in stochastic multi-armed bandits to a continuous arm spectrum, a departure from the limited scope of prior works. The authors' innovative use of Table 1 for succinctly presenting vital outcomes showcases a commendable approach. Adaptation of key concepts from relevant prior studies, along with their application to the continuum arm challenge seems novel. Additionally, the exploration of how the lower and upper bounds are influenced by factors like the corruption budget, zooming dimension, and covering dimension adds substantial value to the paper's findings.

All reviewers are mostly in favor of accepting the submission, but please incorporate all the reviewers' suggestions in the final version of the paper.